# LATENT SPACE LEARNING FOR PDE SYSTEMS WITH COMPLEX BOUNDARY CONDITIONS

## ABSTRACT

Latent space Reduced Order Models (ROMs) in Scientific Machine Learning (SciML) can enhance and accelerate Partial Differential Equation (PDE) simulations. However, they often struggle with complex boundary conditions (BCs) such as time-varying, nonlinear, or state-dependent ones. Current methods for handling BCs in latent space have limitations due to **representation mismatch and projection difficulty**, impacting predictive accuracy and physical consistency. To address this, we introduce BAROM (Boundary-Aware Attention ROM). BAROM integrates: (1) explicit, Reduced Basis Methods-inspired boundary treatment using a modified ansatz and a learnable lifting network for complex BCs; and (2) a non-intrusive, attention-based mechanism, inspired by Galerkin Neural Operators, to learn internal field dynamics within a POD-initialized latent space. Evaluations show BAROM achieves superior accuracy and robustness on benchmark PDEs with diverse complex BCs compared to established SciML approaches.

## 1 INTRODUCTION

Many systems governed by partial differential equations are required to run under tight latency and resource budgets for digital twins, model-predictive control, design and uncertainty quantification, and embedded deployment (LeVeque, 2002; Patankar, 1980; Benner et al., 2015; Quarteroni et al., 2016). Full-order solvers struggle in these settings over long horizons. Reduced-order modeling with latent representations mitigates the cost by advancing a compact state (Antoulas, 2005; Schilders et al., 2008). Classical projection-based methods such as Proper Orthogonal Decomposition and the Reduced Basis Method construct low-dimensional coordinates from high-dimensional fields (Sirovich, 1987; Berkooz et al., 1993; Prud'homme et al., 2002; Rozza et al., 2008; Hesthaven et al., 2016). However, once boundaries become complex, time-varying, or coupled to the interior through feedback or control, reduction becomes fragile. The main issues are representation mismatch caused by global projections that dilute local high-variation boundary effects, instability when forcing physical boundary information into a very small latent state, and limited adaptability of fixed bases (Abbasi et al., 2020; Gunzburger et al., 2007; Knezevic & Patera, 2011). Modified ansatz constructions in Reduced Basis Methods (RBMs) are a known remedy for explicit boundary handling, but intrusive Galerkin dynamics limit scalability in large or complex systems (Abbasi et al., 2020; Tryoen et al., 2010).

Scientific machine learning has broadened the surrogate toolbox, notably through Physics-Informed Neural Networks and Neural Operators (Karniadakis et al., 2021; Rackauckas et al., 2020; Raissi et al., 2019; Lu et al., 2021; Li et al., 2020). Despite many successes, effective boundary imposition remains challenging when boundaries are *complex, nonstationary, or state-dependent*. PINNs rely on loss-weight tuning for soft constraints or handcrafted hard constraints, both of which can be brittle for dynamic or coupled boundaries (Wang et al., 2022b; Cooley et al., 2025; Wang et al., 2021). Operator learners often work well with specific boundary structures such as periodicity, but their robustness under general coupled boundaries is limited (Li et al., 2023; Takeishi & Kalousis, 2021). Among these, reduced-order SciML variants can enhance and accelerate PDE simulations, yet they inherit boundary-handling difficulties from both classical and modern approaches. Specifically, global projections can bias latent spaces toward dominant modes, smoothing out critical local boundary details (Lin et al., 2023). Furthermore, achieving a stable and accurate mapping between latent dynamics and physical boundary constraints remains a significant challenge. Many existing

methods address this by implicitly encoding the entire field, which fails to resolve the underlying issue of explicit boundary enforcement (Li et al., 2024; Zhong & Meidani, 2023; Liu et al., 2024).

In this work, we mainly focus on reduction under two boundary families that are common in practice. The first consists of externally prescribed yet complex and time-varying boundaries. The second involves internal-to-boundary coupling, where boundary values depend on the evolving interior state and possibly a controller. These settings appear in process control, thermal regulation, and transport or flow with actuated boundaries, and they are precisely where many latent ROMs and operator learners become unreliable.

To address these challenges in handling complex boundaries, we propose the Boundary-Aware Attention Reduced Order Model (BAROM), a non-intrusive and boundary-first latent-space ROM that explicitly separates boundary and interior dynamics. BAROM consists of two main components. A learnable lifting network enforces complex non-homogeneous boundaries by construction, while the interior field evolves under homogeneous boundaries in a compact latent state. The evolution of this latent state is governed by a boundary-aware updater that cleanly separates three distinct roles: (1) an attention mechanism captures cross-mode coupling in the latent space, (2) a feed-forward branch models nonlinear self-evolution, and (3) a dedicated branch injects explicit boundary forcing at each time step (Cao, 2021). This explicit decomposition, instead of relying on implicit feature concatenation, provides a strong structural bias that improves long-horizon stability, particularly in closed-loop scenarios.

Using boundary information at the next step is consistent with standard non-homogeneous boundary treatments in classical reduction and is adapted here in a data-driven way (Abbasi et al., 2020). In feedback settings the next-step boundary is computed online from the model's own predicted interior through the specified feedback law, which avoids look-ahead to ground truth. For externally driven settings, competing baselines receive the same accessible temporal and boundary descriptors through a time index and fixed input windows. We document these choices and release code to ensure transparent and fair comparisons.

Compared with strong baselines that treat boundaries implicitly through feature concatenation, adaptive normalization, or soft penalties (Li et al., 2024; Zhong & Meidani, 2023; Li et al., 2023), BAROM enforces boundaries explicitly with a learnable lift and advances the interior with boundary-aware latent dynamics that separate cross-mode coupling, nonlinear self-evolution, and boundary forcing. Our goal is not a general-purpose operator learner, but a boundary-first reduced model for reliable simulation under complex and feedback-coupled boundaries.

Our contributions are as follows:

1. We translate the modified-ansatz concept from classical RBM into a non-intrusive, end-to-end deep learning framework, featuring a learnable lifting network that enforces complex boundary conditions by construction while maintaining efficient latent-space evolution (Abbasi et al., 2020).
2. We design a novel boundary-aware three-branch updater for the latent dynamics, which cleanly separates cross-mode coupling (attention), nonlinear self-dynamics (FFN), and explicit boundary forcing, inspired by operator learning principles (Cao, 2021).
3. We conduct a comprehensive evaluation of BAROM across seven challenging benchmarks with and without internal-to-boundary feedback, demonstrating superior accuracy and stability over state-of-the-art baselines. We provide detailed ablation studies and will release our code and dataset generation protocols to ensure reproducible comparisons.

## 2 PRELIMINARY

**Problem Statement.** We consider physical systems governed by hyperbolic parameterized Partial Differential Equations (PDEs). These equations describe the evolution of a system state $U(x, t; \mu) \in \mathbb{R}^{d_v}$ on a bounded spatial domain $\Omega \subset \mathbb{R}^{d_s}$ over a time interval $\mathcal{T} = (0, T]$. Here, $x \in \Omega$ is the spatial coordinate, $t \in \mathcal{T}$ is time, $\mu$ represents system parameters, and $d_s, d_v$ are the spatial and state vector dimensions, respectively. The governing PDE is generally expressed as:

$$\frac{\partial U}{\partial t} = \mathcal{N}(U, \nabla_x U, \dots; x, t; \mu), \quad \forall (x, t) \in \Omega \times \mathcal{T} \tag{1}$$

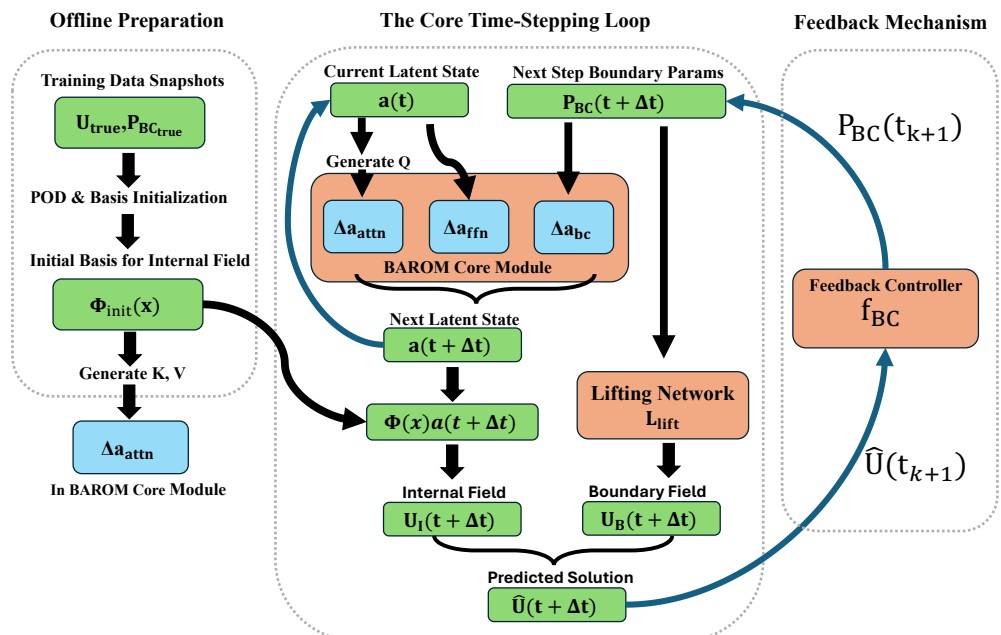

Figure 1: Conceptual diagram of the Boundary-Aware Attention Reduced Order Model (BAROM), highlighting the interaction pathways for boundary control and internal feedback.

where $\mathcal{N}$ is a general nonlinear differential operator defining the system's dynamics, dependent on $U$, its spatial derivatives, $x, t$, and $\mu$. Our focus is on systems with complex Boundary Conditions (BCs) on $\partial\Omega$, categorized as:

*Category One: External Prescribed Complex BCs.* Conditions on $\partial\Omega_1 \subseteq \partial\Omega$ are externally dictated by $P_{ext}(x, t; \mu)$ via a general boundary operator $\mathcal{B}_1$:

$$\mathcal{B}_1(U, \nabla_x U, \ldots; x, t; \mu) = P_{ext}(x, t; \mu), \quad \forall(x, t) \in \partial\Omega_1 \times \mathcal{T} \tag{2}$$

*Category Two: Complex BCs with Internal-Boundary Coupling.* Conditions on $\partial\Omega_2 \subseteq \partial\Omega$ are coupled with the internal state. Here, a boundary operator $\mathcal{B}_2$ imposes values determined by a feedback function $f_{BC}$ which depends on internal states $U(x_s, t; \mu)$ (at locations $x_s \in \Omega \cup \partial\Omega$), potential external controls $u_c(t)$, and parameters $\mu$:

$$\mathcal{B}_2(U, \nabla_x U, \ldots; x, t; \mu) = f_{BC}(U(x_s, t; \mu), u_c(t), \mu), \quad \forall(x, t) \in \partial\Omega_2 \times \mathcal{T} \tag{3}$$

This creates an intrinsic feedback loop where boundary values and internal states are interdependent.

***Objective:*** The primary objective is to develop BAROM, a data-driven ROM for efficiently and accurately simulating PDE systems (Eq. 1) with the complex boundary conditions defined in Categories One and/or Two. BAROM achieves this by jointly learning its three core components: (1) a learnable lifting network ($\mathcal{L}_{\text{lift}}$) for explicit boundary representation, (2) a non-intrusive, boundary-aware latent dynamics model ($\mathcal{F}_{\text{latent}}$) to evolve the internal state, and (3) the spatial basis functions $\Phi(x)$ themselves, which are initialized via POD and refined during training.

The set of all learnable parameters, denoted by $\Theta$, combines the parameters from each of these components:

$$\Theta = \{\Theta_{\mathcal{L}_{\text{lift}}}, \Theta_{\mathcal{F}_{\text{latent}}}, \Theta_{\Phi}\} \tag{4}$$

These parameters are optimized end-to-end by minimizing a loss function, $\mathcal{L}$. This loss function quantifies the discrepancy between BAROM's state predictions, $\hat{U}(\Theta; \cdot)$, and reference solutions, $U_{\text{true}}$ (detailed architecture and learning strategy are presented in Section 3):

$$\min_{\Theta} \mathcal{L}(\hat{U}(\Theta; \cdot), U_{\text{true}}(\cdot)) \tag{5}$$

## 3 METHODOLOGY

Our BAROM framework is a non-intrusive, data-driven realization of the theoretically-grounded principles for handling complex boundary conditions in Reduced Basis Methods (RBM). We begin by establishing the foundational decomposition from classical RBM and then demonstrate how each component of BAROM is designed to learn the operators from the corresponding theoretical governing equations. This structured approach, visualized in Figure 1, allows our model to robustly handle complex boundary dynamics while maintaining computational efficiency.

### 3.1 BAROM FRAMEWORK OVERVIEW

#### 3.1.1 CORE PRINCIPLE: EXPLICIT BOUNDARY-INTERNAL DECOMPOSITION

The core of our approach, inspired by classical RBM for non-homogeneous problems (Abbasi et al., 2020), is the decomposition of the approximate physical solution $\hat{U}(x, t; \mu)$ into a boundary-enforcing component $U_B$ and an internal dynamics component $U_I$. This is expressed through a **modified ansatz**, which is the cornerstone of our framework:

$$\hat{U}(x, t; \mu) = U_B(x, t; P_{BC}(t), \mu) + U_I(x, t; \mu) \tag{6}$$

where the internal field $U_I$ is represented in a low-dimensional latent space using $N$ basis functions $\mathbf{\Phi}(x) = [\phi_1(x), \ldots, \phi_N(x)]$ and a vector of time-dependent latent coefficients $\mathbf{a}(t; \mu) \in \mathbb{R}^N$:

$$U_I(x, t; \mu) = \sum_{i=1}^{N} a_i(t; \mu)\phi_i(x) = \mathbf{\Phi}(x)\mathbf{a}(t; \mu) \tag{7}$$

In this formulation, which directly corresponds to Eq. (7a) in Abbasi et al. (2020):

- **The Boundary Field ($U_B$)** is constructed to satisfy the non-homogeneous physical boundary conditions of the system by construction.
- **The Internal Field ($U_I$)** captures the remaining dynamics. Its basis functions $\mathbf{\Phi}(x)$ are designed to be zero at the boundaries (i.e., they satisfy homogeneous BCs), meaning the entire time-evolution of the internal field is captured by the latent coefficients $\mathbf{a}(t)$.

The key challenge is to define both $U_B$ and the time-evolution of $\mathbf{a}(t)$ in a robust, accurate, and non-intrusive manner. We will now derive BAROM's architecture by directly modeling the governing equations for these two components.

#### 3.1.2 BOUNDARY FIELD ($U_B$) REPRESENTATION

**Classical Formulation.** In the RBM framework presented by Abbasi et al. (2020), the boundary field $U_B$ is a function that "lifts" the boundary values into the full spatial domain. This lifting function is typically a **fixed, manually-defined, and problem-specific interpolant**. For instance, for a 1D problem with domain length $L$, a simple linear interpolation between the boundary values is proposed (see Eq. (35) in Abbasi et al. (2020)):

$$U_B(\hat{U}_{BC}^n) = \left(1 - \frac{X}{L}\right)\hat{U}_{BC}^n|_{x=0} + \frac{X}{L}\hat{U}_{BC}^n|_{x=L} \tag{8}$$

The major drawback of this approach is that a simple interpolant may be a poor approximation of the true boundary field's influence on the interior, and a new function must be hand-crafted for each new problem or geometry.

**BAROM's Learnable Lifting Network.** BAROM replaces this fixed, hand-crafted function with a powerful, general-purpose neural network—the **Lifting Network ($\mathcal{L}_{\text{lift}}$)**, as shown in Figure 1.

$$U_B(x, t; \mu) = \mathcal{L}_{\text{lift}}(P_{BC}(t), \mu; \Theta_{\mathcal{L}_{\text{lift}}})(x) \tag{9}$$

This constitutes a significant advancement over the classical method. Instead of relying on a potentially inaccurate, problem-specific interpolant, BAROM **learns the optimal lifting function** directly from data. The network $\mathcal{L}_{\text{lift}}$ takes the high-level boundary parameters $P_{BC}(t)$ (containing boundary state and control signals) and adaptively generates a physically consistent boundary field $U_B$. This makes the model more general, expressive, and accurate, as it removes the need for manual engineering of the lifting function.

### 3.1.3 INTERNAL FIELD ($U_I$) DYNAMICS

**Classical Formulation: The Theoretical Target.** The core of the dynamics lies in the evolution of the latent coefficients $\mathbf{a}(t)$. In the classical RBM framework, applying a Galerkin projection to the discretized PDE system $U^{n+1} = AU^n + Bw^n$ yields a set of ordinary differential equations (ODEs) for $\mathbf{a}(t)$. For a discrete time step from $n$ to $n+1$, Abbasi et al. (2020) provide the general form of this update in their Eq. (9):

$$\mathbf{a}^{n+1} = \hat{A}_a \mathbf{a}^n + \hat{B}\hat{w}^n + \hat{A}_{BC} U_B^n - \mathbf{\Phi}^T U_B^{n+1} \tag{10}$$

where the operators are defined via intrusive projections: $\hat{A}_a = \mathbf{\Phi}^T A \mathbf{\Phi}$, $\hat{B} = \mathbf{\Phi}^T B$, and $\hat{A}_{BC} = \mathbf{\Phi}^T A$. This equation is our **theoretical target**. Our goal is to create a non-intrusive neural network that learns this exact update structure from data, without ever needing to know the full-order matrices $A$ and $B$.

**BAROM's Non-Intrusive Latent Dynamics Model.** We rewrite the classical update as an additive increment $\Delta \mathbf{a} = \mathbf{a}^{n+1} - \mathbf{a}^n$:

$$\Delta \mathbf{a} = \underbrace{(\hat{A}_a - I)\mathbf{a}^n}_{\substack{\text{Internal Dynamics} \\ \text{(Homogeneous Part)}}} + \underbrace{\left(\hat{B}\hat{w}^n + \hat{A}_{BC} U_B^n - \mathbf{\Phi}^T U_B^{n+1}\right)}_{\text{Boundary Forcing Terms}} \tag{11}$$

BAROM's core module, $\mathcal{F}_{\text{latent}}$, approximates this increment by decomposing it into three specialized components, each modeling a specific part of the theoretical update rule:

$$\mathbf{a}(t_{k+1}) = \mathbf{a}(t_k) + \Delta \mathbf{a}_{\text{attn}} + \Delta \mathbf{a}_{\text{ffn}} + \Delta \mathbf{a}_{\text{bc}} \tag{12}$$

The learnable parameters of this entire module are collectively denoted as $\mathbf{\Theta}_{\mathcal{F}_{\text{latent}}}$.

**Modeling the Internal Dynamics Operator.** The term $(\hat{A}_a - I)\mathbf{a}^n$ represents the autonomous evolution of the internal field. For a general non-linear PDE $\frac{\partial U}{\partial t} = \mathcal{N}(U)$, this term becomes a non-linear vector function of the latent state, $\mathbf{F}_{\text{internal}}(\mathbf{a}^n) = \mathbf{\Phi}^T \mathcal{N}(\mathbf{\Phi}\mathbf{a}^n) - \mathbf{a}^n$. The Universal Approximation Theorem justifies using a neural network to learn this complex, non-linear mapping. We use a structured network that provides a strong inductive bias for latent space dynamics:

- The **Attention mechanism ($\Delta \mathbf{a}_{\text{attn}}$)** models the *cross-modal interactions*, learning a state-dependent coupling operator that approximates how different physical modes (represented by $a_i$ and $a_j$) influence each other.
- The **Feed-Forward Network ($\Delta \mathbf{a}_{\text{ffn}}$)** models the *non-linear self-evolution* of each mode, capturing effects like modal damping or self-reinforcement.

This structured design allows us to learn the internal dynamics operator in a more targeted and physically-interpretable manner:

$$\Delta \mathbf{a}_{\text{attn}} + \Delta \mathbf{a}_{\text{ffn}} \iff \text{learns a structured approximation of} \quad (\mathbf{\Phi}^T A \mathbf{\Phi} - I)\mathbf{a}^n \tag{13}$$

**Modeling the Boundary Forcing Terms.** BAROM's explicit boundary forcing branch, $\Delta \mathbf{a}_{\text{bc}}$, is designed to directly model the entire boundary forcing term from Eq. equation 11:

$$\Delta \mathbf{a}_{\text{bc}} \iff \text{learns an approximation of} \quad (\mathbf{\Phi}^T B)\hat{w}^n + (\mathbf{\Phi}^T A)U_B^n - \mathbf{\Phi}^T U_B^{n+1} \tag{14}$$

We achieve this through a hybrid model that separates the known from the unknown:

$$\Delta \mathbf{a}_{\text{bc}} = \underbrace{g_{bc \to a}(g_{\text{feat}}(P_{BC}(t_{k+1})))}_{\text{Learned Forcing}} - \underbrace{\mathbf{\Phi}(x)^\top U_B(x, t_{k+1})}_{\text{Theoretical Correction}} \tag{15}$$

The mapping to the classical theory is direct and rigorous:

- The term $-\mathbf{\Phi}(x)^\top U_B(x, t_{k+1})$ is a **direct, non-intrusive implementation of the crucial theoretical correction term** $-\mathbf{\Phi}^T U_B^{n+1}$. This term is known to be essential for stability (Abbasi et al., 2020), and by including it directly, we imbue our model with a strong, physically-grounded inductive bias.
- The learned component, $g_{bc \to a}(\dots)$, is a neural network that serves as a universal approximator for the remaining complex forcing terms, which are functions of the current-step boundary conditions. It learns the combined effect of these terms, $(\mathbf{\Phi}^T B)\hat{w}^n + (\mathbf{\Phi}^T A)U_B^n$, by mapping the high-level boundary parameters $P_{BC}$ to the resulting latent space increment. This bypasses the need to explicitly compute the intrusive and unknown matrices $A$ and $B$.

### 3.1.4 THE CORE TIME-STEPPING LOOP

The complete predictive process of BAROM, as depicted in Figure 1, integrates these theoretically-grounded components into a single, efficient time-stepping loop. For each step from $t_k$ to $t_{k+1}$:

1. **Inputs:** The model receives the current latent state $\mathbf{a}(t_k)$ and the parameters for the *next* boundary condition, $P_{BC}(t_{k+1})$. For feedback-controlled systems, $P_{BC}(t_{k+1})$ is computed from the model's own prior prediction $\hat{U}(t_k)$, as shown by the "Feedback Mechanism" loop in Figure 1.
2. **Latent Update:** The "BAROM Core Module" computes the three increments $(\Delta\mathbf{a}_{\text{attn}}, \Delta\mathbf{a}_{\text{ffn}}, \Delta\mathbf{a}_{\text{bc}})$ as derived above. These are added to $\mathbf{a}(t_k)$ to produce the next latent state, $\mathbf{a}(t_{k+1})$.
3. **Physical Reconstruction:** The final predicted solution $\hat{U}(t_{k+1})$ is reconstructed using the modified ansatz (Eq. equation 6):
   - The new latent state $\mathbf{a}(t_{k+1})$ is linearly combined with the basis functions $\boldsymbol{\Phi}$ to form the Internal Field $U_I(t_{k+1})$.
   - The Lifting Network $\mathcal{L}_{\text{lift}}$ uses $P_{BC}(t_{k+1})$ to generate the corresponding Boundary Field $U_B(t_{k+1})$.
   - The two fields are summed: $\hat{U}(t_{k+1}) = U_B(t_{k+1}) + U_I(t_{k+1})$.

The final solution is reconstructed using the learnable basis functions (parameterized by $\boldsymbol{\Theta_\Phi}$). This entire process is summarized in Algorithm 1. By structuring our architecture as a direct, data-driven analogue of the classical RBM theory, BAROM combines the robustness of established physical principles with the expressive power and non-intrusive nature of deep learning.

### 3.1.5 LOSS FUNCTION AND OPTIMIZATION

The BAROM model parameters $\Theta = \{\boldsymbol{\Theta}_{\mathcal{L}_{\text{lift}}}, \boldsymbol{\Theta}_{\mathcal{F}_{\text{latent}}}, \boldsymbol{\Theta_\Phi}\}$ are trained end-to-end by minimizing a composite loss function $\mathcal{L}(\Theta)$ on high-fidelity simulation data $\mathcal{D}_{\text{train}}$. The primary term is the spatio-temporal reconstruction loss $\mathcal{L}_{\text{recon}}$, which measures the discrepancy between the predicted solution $\hat{U}$ and the true solution $U_{\text{true}}$:

$$\mathcal{L}_{\text{recon}} = \mathbb{E}_{(U_{\text{true}}, P_{BC}) \sim \mathcal{D}_{\text{train}}} \left[ \frac{1}{T_{\text{sim}}} \int_0^{T_{\text{sim}}} \frac{1}{|\Omega|} \int_\Omega \|\hat{U}(x,t;\mu) - U_{\text{true}}(x,t;\mu)\|_2^2 dx dt \right] \quad (16)$$

To improve the physical properties of the learned basis functions $\boldsymbol{\Phi}$, we introduce two regularization terms. The final composite loss function is:

$$\mathcal{L}(\Theta) = \mathcal{L}_{\text{recon}} + \lambda_{\text{orth}} \mathcal{L}_{\text{orth}}(\boldsymbol{\Phi}) + \lambda_{\text{bc}} \mathcal{L}_{\text{bc\_pen}}(\boldsymbol{\Phi}) \quad (17)$$

The weights $\lambda_{\text{orth}}$ and $\lambda_{\text{bc}}$ are hyperparameters that balance the trade-off between reconstruction accuracy and the enforcement of physical priors on the basis functions. These values were determined empirically through a validation study. We provide a detailed sensitivity analysis of these weights in Appendix F, which supports the choice of values used in our main experiments.

The regularization terms are defined as follows:

- $\mathcal{L}_{\text{orth}}(\boldsymbol{\Phi}) = \|\boldsymbol{\Phi}^T\boldsymbol{\Phi} - \mathbf{I}\|_F^2$: An orthogonality penalty that encourages the basis functions to be orthonormal.
- $\mathcal{L}_{\text{bc\_pen}}(\boldsymbol{\Phi}) = \|\boldsymbol{\Phi}|_{\partial\Omega}\|_F^2$: A boundary penalty on the basis functions.

It is crucial to clarify the role of $\mathcal{L}_{\text{bc\_pen}}(\boldsymbol{\Phi})$. This loss term serves as a regularization penalty and does not enforce the physical boundary conditions of the overall system. The physical BCs on the final solution $\hat{U}$ are satisfied **by construction** via the lifting network's output $U_B$. Instead, $\mathcal{L}_{\text{bc\_pen}}$ acts as a soft constraint specifically on the **learnable basis functions $\boldsymbol{\Phi}$** of the *internal field $U_I$*. By encouraging these basis functions themselves to approach zero on the boundary ($\boldsymbol{\Phi}|_{\partial\Omega} \approx 0$), this term helps ensure that the internal solution component $U_I = \boldsymbol{\Phi}\mathbf{a}$ correctly satisfies its required homogeneous boundary conditions. This aligns with the explicit decomposition principle central to our framework and aids in stabilizing the training of the basis functions.

---

**Algorithm 1** BAROM Predictive Step for $\hat{U}(x, t_{k+1})$

---

1: **Input:** Current latent state $\mathbf{a}(t_k)$, learnable bases $\mathbf{\Phi}$, boundary parameters for next step $P_{BC}(t_{k+1})$, and model parameters $\Theta = \{\mathbf{\Theta}_{\mathcal{L}_{\text{lift}}}, \mathbf{\Theta}_{\mathcal{F}_{\text{latent}}}, \mathbf{\Theta}_{\mathbf{\Phi}}\}$.
2: Evolve latent coefficients using $\mathcal{F}_{\text{latent}}$.
3: Compute $\Delta \mathbf{a}_{\text{attn}}$ via attention using $\mathbf{a}(t_k)$ and $\mathbf{\Phi}$.
4: Compute $\Delta \mathbf{a}_{\text{ffn}}$ via FFN using $\mathbf{a}(t_k)$.
5: Compute $\Delta \mathbf{a}_{\text{bc}}$ by processing $P_{BC}(t_{k+1})$.
6: $\mathbf{a}(t_{k+1}) \leftarrow \mathbf{a}(t_k) + \Delta \mathbf{a}_{\text{attn}} + \Delta \mathbf{a}_{\text{ffn}} + \Delta \mathbf{a}_{\text{bc}}$.              ▷ Construct solution components
7: $U_B(x, t_{k+1}) \leftarrow \mathcal{L}_{\text{lift}}(P_{BC}(t_{k+1}), \mu; \Theta_{\text{lift}})(x)$.
8: $U_I(x, t_{k+1}) \leftarrow \mathbf{\Phi}(x)\mathbf{a}(t_{k+1})$.
9: $\hat{U}(x, t_{k+1}) \leftarrow U_B(x, t_{k+1}) + U_I(x, t_{k+1})$.
10: **Output:** Next latent state $\mathbf{a}(t_{k+1})$, predicted physical state $\hat{U}(x, t_{k+1})$.
  *Note: Initial latent state $\mathbf{a}(t_0)$ is obtained by projecting the initial physical state $U(x, t_0)$ onto $\mathbf{\Phi}(x)$ after subtracting the initial boundary field $U_B(x, t_0; P_{BC}(t_0))$.*

---

## 4 EXPERIMENTS

We conduct a series of experiments to validate BAROM's performance on diverse PDE systems featuring complex boundary conditions (BCs). Our evaluation is structured around key research questions to assess the model's effectiveness, architectural contributions, and robustness compared to state-of-the-art baselines.

### 4.1 EXPERIMENTAL SETUP

**Datasets.** We evaluate BAROM on seven PDE benchmarks across two challenging categories: (1) systems with externally prescribed, complex time-varying BCs (e.g., Advection-Reaction, Burgers', Euler, Darcy Flow), and (2) systems with tightly coupled internal-boundary feedback and control (e.g., RDFNF, Heat_NF, Convdiff). The full mathematical formulations, parameter distributions, and feedback laws for all datasets are provided in Appendix A.

**Baselines and Metrics.** We benchmark BAROM against a suite of representative SciML models: SPFNO (Liu et al., 2023), BENO (Wang et al., 2024), LNS-AE (Li et al., 2024), LNO (Wang & Wang, 2024), POD-DL-ROM (Fresca & Manzoni, 2022), GNOT (Hao et al., 2023), and Unisolver (Zhou et al., 2024). Performance is primarily assessed using Mean Squared Error (MSE), Relative $L_2$ Error (RelErr), and Maximum Absolute Error (MaxErr), averaged over test samples.

**Fairness of Comparison.** Our primary model, **BAROM_ExpBC**, leverages next-step boundary information, a core architectural feature motivated by classical RBM theory for non-homogeneous BCs. To ensure a direct, apples-to-apples comparison, we also evaluate **BAROM_ImpBC**, which uses a standard auto-regressive setup identical to the baselines, relying only on current-step information. A detailed discussion on this setup is available in Appendix C.3.

### 4.2 RESULTS

**RQ1: How does BAROM perform on challenging PDE systems with complex, feedback-controlled boundary conditions compared to state-of-the-art baselines?** **Answer:** BAROM demonstrates substantially superior accuracy and robustness. As shown in Table 1, our full model, **BAROM_ExpBC**, consistently outperforms all baseline models across all three feedback-controlled datasets and time horizons. The performance gains are significant; for instance, on the Heat_NF dataset at $T = 1.5$, BAROM_ExpBC reduces MSE by more than 79% compared to the next best stable model, SPFNO.

Crucially, our fair-comparison variant, **BAROM_ImpBC**, also consistently and substantially outperforms all baseline models. This result unequivocally demonstrates that the core architectural innovations of BAROM, the learnable lifting network and the boundary-aware attention mecha-

nism—provide a significant performance advantage on their own, independent of the theoretically-grounded next-step BC forcing.

**RQ2: What are the contributions of BAROM's key architectural components to its overall performance?** **Answer:** Our ablation studies, detailed in Appendix C.2, systematically validate our core design principles on the RDFNF dataset. The key findings are:

- **Learnable Lifting is Critical:** The learnable lifting network ($\mathcal{L}_{\text{lift}}$) leads to a substantial improvement in prediction accuracy compared to a fixed, non-learnable lifting function, underscoring the importance of an adaptive mechanism for representing the boundary field.
- **Explicit BC Forcing is Crucial:** Explicitly providing boundary parameters to the latent dynamics model significantly reduces error for all architectures. The Boundary-Aware Attention (BAA) mechanism, when combined with this explicit input, achieves the lowest overall error.
- **End-to-End Training is Robust:** Our framework's end-to-end training is powerful enough to discover a more effective basis for predicting dynamics from a random start than the one prescribed by a static, energy-based POD initialization.

**RQ3: How does BAROM perform on systems with externally driven (non-feedback) complex boundary conditions?** **Answer:** BAROM remains highly effective and robust. On four canonical PDE systems with complex, externally prescribed BCs, BAROM demonstrates stable and accurate performance, surpassing most baselines, particularly in long-term extrapolation scenarios where competing models like SPFNO and BENO encountered instabilities. For these challenging hyperbolic problems, BAROM maintained stability and delivered superior accuracy in longer-term predictions. Detailed quantitative results for these experiments are presented in Appendix C.1, Table 2.

**Qualitative Analysis.** These quantitative findings are further corroborated by a detailed visual analysis in Appendix B, which provides a qualitative comparison of the predicted solution fields against the ground truth for representative test cases.

Table 1: Performance comparison on PDE systems with internal feedback and Boundary control at $T = 1.5, nt = 225, T = 1.75, nt = 265$ and $T = 2.0, nt = 300$. Metrics are averaged over 1000 test samples. Best results are in **bold**, second best resutls are in underline.

| T=1.5 | Heat_NF | | | RDFNF | | | Convdiff | | |
| Model | MSE(e-2) | RelErr(e-1) | MaxErr(e-1) | MSE(e-2) | RelErr(e-1) | MaxErr(e-1) | MSE(e-2) | RelErr(e-1) | MaxErr(e-1) |
|---|---|---|---|---|---|---|---|---|---|
| BAROM_ImpBC (ours) | 2.603 | 3.456 | 8.678 | 2.459 | 2.818 | 9.377 | 3.618 | 1.940 | 4.982 |
| BAROM_ExpBC (ours) | **1.181** | **2.280** | **8.115** | **1.095** | **1.883** | **7.392** | **2.081** | **1.496** | **4.703** |
| SPFNO | 5.806 | 5.079 | 15.26 | 5.107 | 4.119 | 16.04 | 8.684 | 3.145 | 14.19 |
| BENO | 474.5 | 46.43 | 51.23 | 231.6 | 88.19 | 92.12 | 10.26 | 3.32 | 12.32 |
| LNS-AE | 11.15 | 6.721 | 18.52 | 16.77 | 7.142 | 16.99 | 10.48 | 3.206 | 1.050 |
| LNO | 16.62 | 8.023 | 24.43 | 5.521 | 4.246 | 15.04 | 34.28 | 5.314 | 22.43 |
| POD-DL | 9.958 | 6.649 | 15.95 | 5.193 | 4.197 | 16.04 | 4.871 | 2.313 | 14.19 |
| UNISOLVER | 15.91 | 8.729 | 17.59 | 9.888 | 5.194 | 19.54 | 9.691 | 3.223 | 11.03 |
| GNOT | 174.3 | 25.86 | 42.55 | 72.732 | 15.338 | 53.657 | 3.3072 | 1.9 | 8.005 |

| T=1.75 | Heat_NF | | | RDFNF | | | Convdiff | | |
| Model | MSE(e-2) | RelErr(e-1) | MaxErr(e-1) | MSE(e-2) | RelErr(e-1) | MaxErr(e-1) | MSE(e-2) | RelErr(e-1) | MaxErr(e-1) |
|---|---|---|---|---|---|---|---|---|---|
| BAROM_ImpBC (ours) | 2.674 | 3.595 | 8.809 | 2.528 | 2.723 | 9.521 | 3.877 | 1.938 | 5.182 |
| BAROM_ExpBC (ours) | **1.199** | **2.348** | **8.198** | **1.168** | **1.862** | **7.630** | **2.234** | **1.487** | **4.930** |
| SPFNO | 6.038 | 5.314 | 15.75 | 5.432 | 4.046 | 16.58 | 8.921 | 3.111 | 1.419 |
| BENO | 518.11 | 57.12 | 56.43 | 263.5 | 89.43 | 92.41 | 12.34 | 3.536 | 13.45 |
| LNS-AE | 12.38 | 7.318 | 19.45 | 18.87 | 7.210 | 16.34 | 1.232 | 3.378 | 11.11 |
| LNO | 23.16 | 9.834 | 29.51 | 6.208 | 4.304 | 15.84 | 45.81 | 5.92 | 26.33 |
| POD-DL | 9.337 | 6.605 | 16.35 | 5.033 | 3.937 | 16.58 | 4.565 | 2.178 | 14.19 |
| UNISOLVER | 16.71 | 8.77 | 16.78 | 10.12 | 5.494 | 19.12 | 10.01 | 3.215 | 11.22 |
| GNOT | 193.10 | 28.21 | 43.789 | 88.0 | 16.15 | 55.47 | 3.3718 | 1.865 | 8.101 |

## 5 RELATED WORK

Latent space methods, from classical Reduced Basis Methods (RBM) (Antoulas, 2005; Prud'homme et al., 2002; Hesthaven et al., 2016) to modern Autoencoders (AE) (Lee & Carlberg, 2020; Hinton & Salakhutdinov, 2006), accelerate PDE simulations via low-dimensional representations (Wiewel et al., 2019; Champion et al., 2019). However, accurately imposing complex Boundary Conditions

| T=2 | Heat_NF | | | RDFNF | | | Convdiff | | |
|---|---|---|---|---|---|---|---|---|---|
| Model | MSE(e-2) | RelErr(e-1) | MaxErr(e-1) | MSE(e-2) | RelErr(e-1) | MaxErr(e-1) | MSE(e-2) | RelErr(e-1) | MaxErr(e-1) |
| BAROM_ImpBC (ours) | 2.826 | 3.751 | 9.264 | 2.616 | 2.648 | 9.769 | 4.432 | 2.005 | 5.646 |
| BAROM_ExpBC (ours) | **1.264** | **2.440** | **8.484** | **1.232** | **1.833** | **7.956** | **2.379** | **1.424** | **5.099** |
| SPFNO | 6.458 | 5.5711 | 16.75 | 5.814 | 3.999 | 17.13 | 9.252 | 3.103 | 14.19 |
| BENO | 551.3 | 5.228 | 57.21 | 288.5 | 89.41 | 94.03 | 13.93 | 3.646 | 13.71 |
| LNS-AE | 13.27 | 7.744 | 20.12 | 20.90 | 7.241 | 16.65 | 13.86 | 3.536 | 11.57 |
| LNO | 31.03 | 11.31 | 33.87 | 6.821 | 4.328 | 16.63 | 59.68 | 6.611 | 31.310 |
| POD-DL | 9.066 | 6.601 | 17.21 | 5.101 | 3.794 | 17.13 | 4.938 | 2.210 | 14.19 |
| UNISOLVER | 17.95 | 8.848 | 16.41 | 10.47 | 5.869 | 18.71 | 10.51 | 3.237 | 11.32 |
| GNOT | 208.1 | 30.01 | 45.01 | 99.61 | 16.48 | 56.97 | 3.582 | 1.876 | 8.232 |

(BCs) remains a significant challenge. While simple BCs are manageable (e.g., (Ohlberger & Rave, 2013; Reiss et al., 2018)), dimensionality reduction often compromises critical boundary information in complex scenarios (Abbasi et al., 2020), and many latent models lack explicit boundary interaction.

Current SciML approaches integrating latent spaces often show limited robust coupling between latent dynamics and physical boundary variations. Many AE-based models or latent solvers like LNS (Li et al., 2024) implicitly encode the entire field, risking smoothed or lost local boundary details (Abbasi et al., 2020). Without explicit real-time BC input, predictions under dynamic boundaries can be inaccurate. Some works incorporate Physics-Informed Neural Network (PINN) concepts (Raissi et al., 2019), for instance, in combinations with Variational Autoencoders (Zhong & Meidani, 2023). These typically use soft constraints for BCs, whose effectiveness depends on loss weighting and may not fully resolve decoupling issues (Wang et al., 2022a). Neural Operators (NOs), including DeepONet (Lu et al., 2021) and FNO (Li et al., 2020), often target specific BCs, and their capability to robustly handle diverse complex BCs within their latent dynamics is not guaranteed.

In contrast, traditional RBMs offer valuable strategies. Notably, Abbasi et al. (2020) (Abbasi et al., 2020) demonstrated an approach for explicit boundary effect separation using modified solution structures and specialized basis functions for varying and nonlinear BCs. This principle of explicit, representation-level BC handling provides key inspiration for developing more robust SciML methods to address boundary challenges within latent space frameworks.

# 6 CONCLUSIONS

We introduced the Boundary-Aware Attention Reduced Order Model (BAROM), a novel framework adept at simulating Partial Differential Equation systems with complex, dynamic, and feedback-dependent boundary conditions. BAROM uniquely synergizes a Reduced Basis Method inspired explicit boundary treatment with a data-driven, boundary-aware latent dynamics model. To directly address the challenge of achieving an effective latent representation for intricate BCs, BAROM employs a learnable lifting network for inhomogeneous conditions and specialized learnable bases for the homogeneous internal field. Furthermore, to ensure accurate boundary enforcement and maintain physical consistency within the latent space, the model non-intrusively evolves the system's low-dimensional state representation using attention mechanisms and direct integration of boundary parameter information. Extensive evaluations confirmed BAROM's superior accuracy, robustness, and long-term prediction capabilities over state-of-the-art baselines, particularly for challenging feedback-controlled systems. Ablation studies further validated its core design principles. BAROM thus offers an effective Scientific Machine Learning approach for complex physical simulations, opening promising avenues for future architectural enhancements and broader applications in challenging scientific and engineering domains.

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

## A DATASET DETAILS

Our PDE datasets for category one problem2 include four canonical PDE problems—Advection-Reaction, Isothermal Euler, Burgers', and Darcy equations—with nonlinear, time-varying, and controlled boundary conditions. Each dataset contains 10,000 samples generated using NumPy and SciPy, with spatial discretization set at $nx = 128$ and temporal discretization at $nt = 600$ over a simulation time horizon $T = 2$. The Advection dataset solves a scalar linear PDE with controlled sinusoidal boundary conditions. The Euler dataset involves coupled nonlinear PDEs with nonlinear boundary conditions ensuring positive densities. The Burgers' dataset applies Crank-Nicolson and Lax-Friedrichs schemes to handle advection and diffusion separately, enforcing stability through smoothed boundary conditions. The Darcy dataset is generated via solving a steady-state elliptic PDE with randomized log-normal permeability fields and time-dependent Dirichlet boundary conditions. Data samples include state solutions, boundary states, and detailed parameterizations, ensuring comprehensive and varied scenarios for benchmarking models. We also construct datasets for category two problem2 covering diverse PDE scenarios with integral and nonlinear boundary feedback controls, focusing on four distinct PDE types: Heat Equation with Delayed Integral Feedback, Reaction-Diffusion Equation with Neumann Boundary Integral Feedback, Heat Equation with Nonlinear Feedback Gain, and Convection-Diffusion Equation with Integral Boundary Control. Each dataset comprises 5,000 samples, discretized spatially at $nx = 64$ and temporally at $nt = 300$ over a simulation interval of $T = 2.0$. Boundary conditions incorporate complex, nonlinear time-varying signals generated via composite sinusoidal and polynomial-like forms. Control actions at boundaries are realized through integral and nonlinear feedback laws, adding realistic control dynamics. Each sample includes PDE solution fields, boundary states, control signals, and comprehensive parameterizations, providing extensive variability for rigorous model evaluation and benchmarking.

This appendix also provides the complete mathematical formulations for the seven PDE benchmark datasets used in our experiments.

### A.1 PDE SYSTEMS FOR SECTION 2: EXTERNALLY PRESCRIBED BCS

#### A.1.1 1D ADVECTION-REACTION

**Governing Equation:**

$$\frac{\partial u}{\partial t} + c\frac{\partial u}{\partial x} = ru, \quad x \in [0,1], t \in [0,2] \tag{18}$$

with advection speed $c = 1.0$ and reaction rate $r = 0.5$.

**Initial Condition:** A Gaussian pulse with randomized amplitude.

$$u(x,0) = \mu_1 \exp\left(-\frac{(x-0.5)^2}{0.1}\right), \quad \mu_1 \sim \mathcal{U}(0.8, 1.2) \tag{19}$$

**Boundary Conditions (Dirichlet):**

$$u(0,t) = A\sin(\pi t) + B\cos(\pi t) + A\sin^3(\pi t) + E\sin(3\pi t) + F\cos(3\pi t) + u_{c1}(t)$$

$$u(1,t) = C\sin(\pi t) + D\cos(2\pi t) + D\cos^3(\pi t) + \tanh(\sin(\pi t)) + Ge^{-\frac{(t-1)^2}{0.04}}$$

where coefficients $A, B, \ldots, G \sim \mathcal{U}(0.5, 1.5)$. The control signal $u_{c1}(t)$ is a smoothed random step function.

#### A.1.2 1D ISOTHERMAL EULER

**Governing Equations:** A system for density $\rho$ and velocity $u$.

$$\frac{\partial \rho}{\partial t} + \frac{\partial(\rho u)}{\partial x} = 0$$

$$\frac{\partial(\rho u)}{\partial t} + \frac{\partial(\rho u^2 + \rho)}{\partial x} = 0 \tag{20}$$

**Initial Condition:**    Sinusoidal perturbations on a mean state.

$$\rho(x,0) = \rho_{\text{mean}} + \rho_{\text{amp}}\sin(k\pi x), \quad \rho_{\text{mean}} \sim \mathcal{U}(0.8, 1.2), \rho_{\text{amp}} \sim \mathcal{U}(0, 0.1)$$
$$u(x,0) = u_{\text{mean}} + u_{\text{amp}}\cos(k\pi x), \quad u_{\text{mean}} \sim \mathcal{U}(-0.2, 0.2), u_{\text{amp}} \sim \mathcal{U}(0, 0.05)$$

with wavenumber $k \in \{1, 2\}$.

**Boundary Conditions (Dirichlet):**

$$\rho(0,t) = \rho_{L,\text{base}} + \rho_{L,\text{amp}}\sin(\omega_{\rho_L}t) \quad u(0,t) = u_{L,\text{base}} + u_{c1}(t)$$
$$\rho(1,t) = \rho_{R,\text{base}} + \rho_{R,\text{amp}}\cos(\omega_{\rho_R}t) \quad u(1,t) = u_{R,\text{base}} - \kappa\tanh(5(\rho(1,t) - \rho_{R,\text{base}})) + u_{c2}(t)$$

where base values, amplitudes, frequencies, and the non-linear coefficient $\kappa$ are randomized. Controls $u_{c1}(t), u_{c2}(t)$ are smoothed random step functions.

### A.1.3    1D VISCOUS BURGERS'

**Governing Equation:**

$$\frac{\partial u}{\partial t} + u\frac{\partial u}{\partial x} = \nu\frac{\partial^2 u}{\partial x^2} \tag{21}$$

with viscosity $\nu \sim \mathcal{U}(0.5, 1.5) \cdot (0.01/\pi)$.

**Initial Condition:**

$$u(x,0) = \alpha\sin(k\pi x), \quad \alpha \sim \mathcal{U}(0.5, 1.5), k \in \{1, 2, 3\} \tag{22}$$

**Boundary Conditions:**

- **Left (Dirichlet,** $x = 0$**):** $u(0,t) = A_l[1 + 0.5\sin(1.5\omega_l t)]\sin(\omega_l t) + B_l u(x_1, t)^2 + u_{c1}(t)$
- **Right (Robin,** $x = 1$**):** $\nu u_x(1,t) + \beta_r(u(1,t)^3 - u_{\text{ref}}^3) = u_{c2}(t)$

where all parameters are randomized. The term $u(x_1, t)^2$ denotes a state dependency on the first interior grid point.

### A.1.4    2D DARCY FLOW

**Governing Equation:**    A steady-state equation for pressure $p(x)$ solved at multiple time instances.

$$-\nabla \cdot (k(x)\nabla p(x)) = f(x) \tag{23}$$

where the permeability $k(x)$ is a log-normal random field and source $f(x) = 0$.

**Boundary Conditions (Dirichlet):**    For each time snapshot $t_n$, a new pair of BCs is applied.

$$p(0, y, t_n) = P_L(t_n) \qquad\qquad p(1, y, t_n) = P_R(t_n)$$
$$p(x, 0, t_n) = P_B(t_n) \qquad\qquad p(x, 1, t_n) = P_T(t_n)$$

where each boundary value $P(t_n)$ follows a complex, non-linear sinusoidal function of time, with randomized parameters.

## A.2    PDE SYSTEMS FOR SECTION 1: INTERNAL FEEDBACK AND CONTROL

### A.2.1    CONVECTION-DIFFUSION WITH INTEGRAL FEEDBACK

**Governing Equation:**

$$\frac{\partial u}{\partial t} + a\frac{\partial u}{\partial x} = D\frac{\partial^2 u}{\partial x^2} \tag{24}$$

**Initial Condition:**    $u(x,0) = \sin(\pi x) + 0.1 \cdot \mathcal{N}(x)$

**Boundary Conditions (Dirichlet with Integral Control):**

$$u(0,t) = r_{\text{left}}(t) + c_{\text{left}}(t), \quad \text{where } c_{\text{left}}(t) = K_I \int_0^t [r_{\text{left}}(\tau) - u(0,\tau)]d\tau$$

$$u(1,t) = r_{\text{right}}(t) + c_{\text{right}}(t), \quad \text{where } c_{\text{right}}(t) = K_I \int_0^t [r_{\text{right}}(\tau) - u(1,\tau)]d\tau$$

where $r(t)$ are time-varying reference signals and $K_I$ is the integral gain.

### A.2.2 REACTION-DIFFUSION WITH NEUMANN INTEGRAL FEEDBACK

**Governing Equation:**

$$\frac{\partial u}{\partial t} = \alpha \frac{\partial^2 u}{\partial x^2} + \mu u(1-u) \tag{25}$$

**Initial Condition:** $u(x,0) = A_0 \sin^2(\pi f_0 x)$

**Boundary Conditions:**

- **Left (Dirichlet, $x = 0$):** $u(0,t) = g_0(t) + c_0(t)$
- **Right (Neumann with Integral Feedback, $x = 1$):** $\alpha \frac{\partial u}{\partial x}(1,t) = g_L(t) + K_{\text{fb}} \int_0^1 u(x,t)dx + c_L(t)$

where $g(t)$ are base signals, $c(t)$ are external controls, and $K_{\text{fb}}$ is the feedback gain.

### A.2.3 HEAT EQUATION WITH NON-LINEAR FEEDBACK GAIN

**Governing Equation:**

$$\frac{\partial u}{\partial t} = \alpha \frac{\partial^2 u}{\partial x^2} \tag{26}$$

**Initial Condition:** $u(x,0) = A_0 \sin(\pi f_0 x)$

**Boundary Conditions (Dirichlet with Non-linear Feedback):**

$$u(0,t) = g_0(t) + c_0(t)$$

$$u(1,t) = g_L(t) + F\left(\int_0^1 u(x,t)dx\right) + c_L(t)$$

where the feedback function $F$ is a non-linear (quadratic) function of the spatially integrated state: $F(s) = K_1 s + K_2 s^2$.

## B VISUALIZATION ANALYSIS

To complement the quantitative results, this section provides a qualitative visual analysis of BAROM's predictive capabilities compared to baseline models on representative test cases. We present spatio-temporal contour plots of the predicted solution fields against the ground truth.

Figure 3 illustrates the performance on the 1D Euler equations, showcasing predictions up to $T = 1.0$ (training horizon) and extrapolation to $T = 2.0$. BAROM maintains solution structure and accurately captures key dynamic features, such as shock propagation and contact discontinuities, even in the extrapolation regime. In contrast, several baseline models exhibit either excessive numerical diffusion, oscillations, or a failure to preserve essential wave characteristics, particularly at later time points. For instance, LNO and POD-DL-ROM show significant deviations in wave patterns and amplitudes when extrapolating, while BENO struggles with the sharp features inherent in hyperbolic systems. LNS-AE, while more stable, also shows some loss of definition in shock fronts compared to BAROM.

For the feedback-controlled systems, such as the Convection-Diffusion equation shown in Figure 4 and Figure 5 , BAROM's predictions remain highly consistent with the ground truth across different

time horizons. The model accurately captures the solution's response to the integral boundary control, maintaining stability and physical plausibility. Some baselines, like SPFNO, show reasonable adherence but with noticeable discrepancies, while others like LNO and BENO demonstrate more significant deviations or instabilities, especially in the extrapolation case ($T = 2.0$). Similar observations are made for the Heat Equation with Delayed Integral Feedback (Figure 6 and Figure 7), where BAROM effectively models the complex interplay between internal dynamics and boundary feedback. The qualitative results for the Burgers' equation further underscore BAROM's capability to handle nonlinear dynamics and shock formation accurately over time.

Overall, these visualizations corroborate the quantitative findings, highlighting BAROM's enhanced robustness and accuracy in predicting complex PDE solutions under diverse and challenging boundary conditions, particularly in scenarios involving feedback control and long-term temporal evolution.

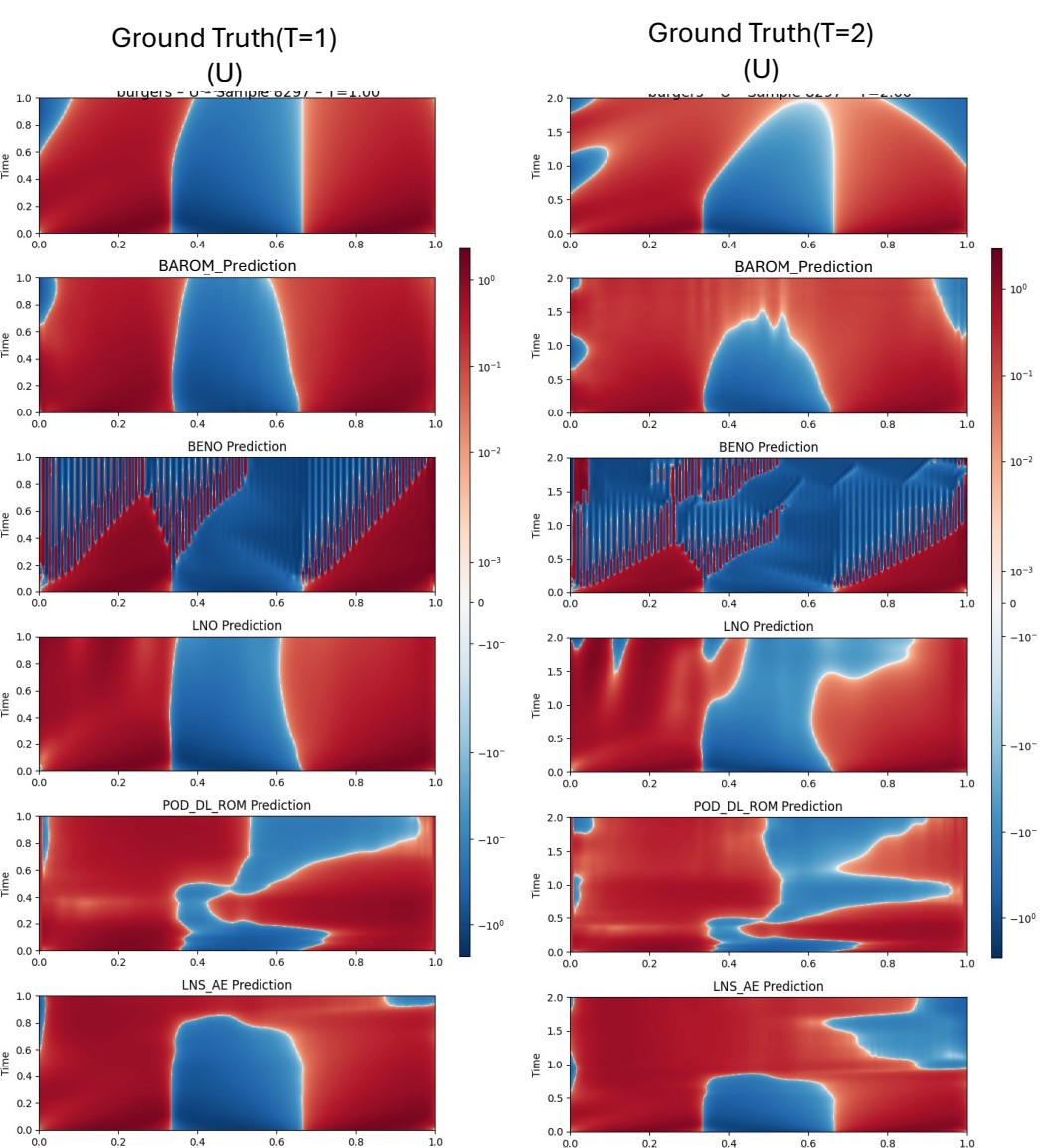

Figure 2: Visualization of prediction Burgers data at $T_{\mathrm{horizon}} = 1$ and 2.

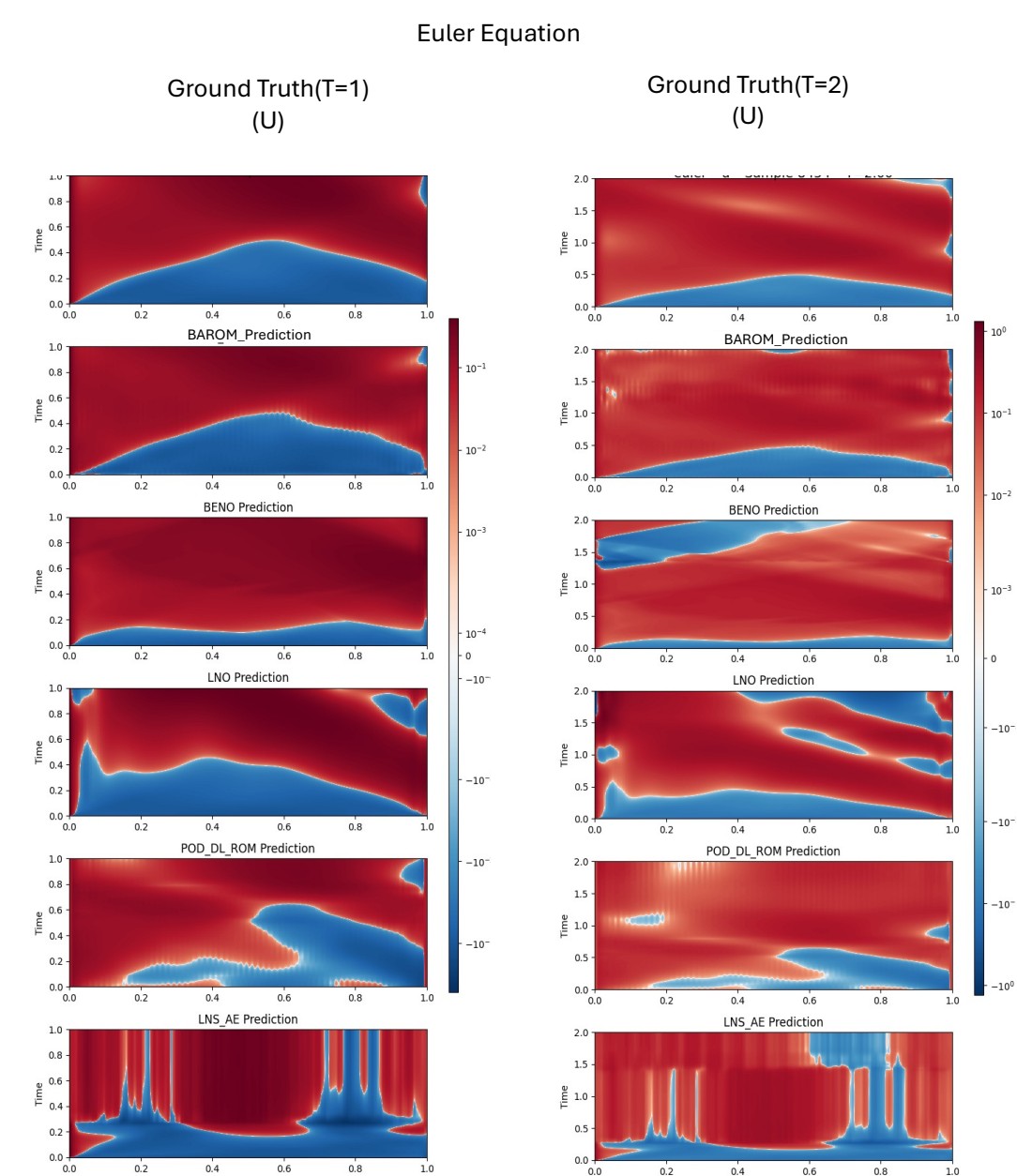

Figure 3: Visualization of prediction Euler Equation data at $T_{\text{horizon}} = 1$ and 2.

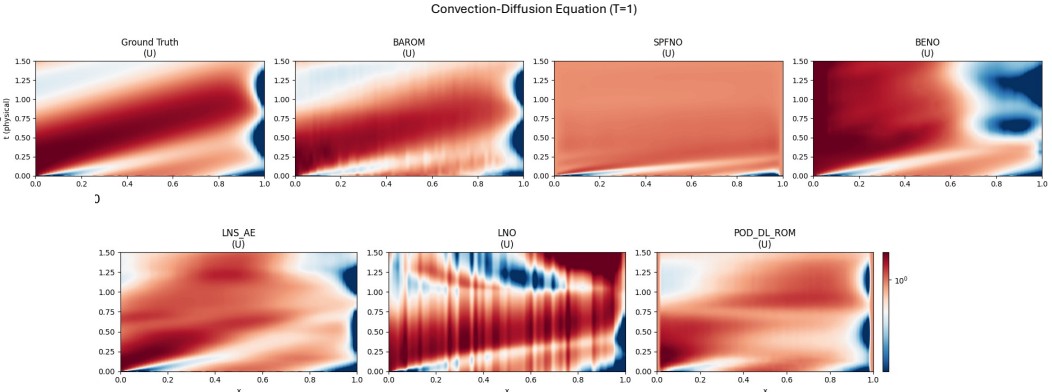

Figure 4: Visualization of prediction Convection–Diffusion data at $T_{\text{horizon}} = 1$.

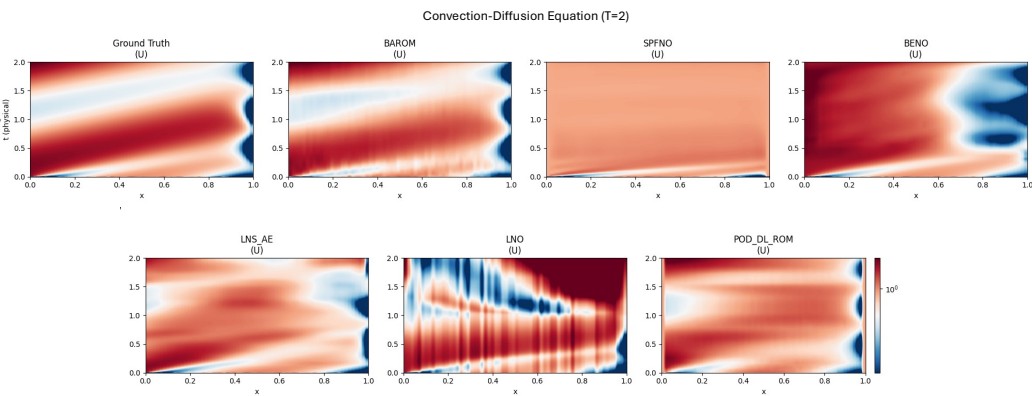

Figure 5: Visualization of prediction Convection–Diffusion data at $T_{\text{horizon}} = 2$.

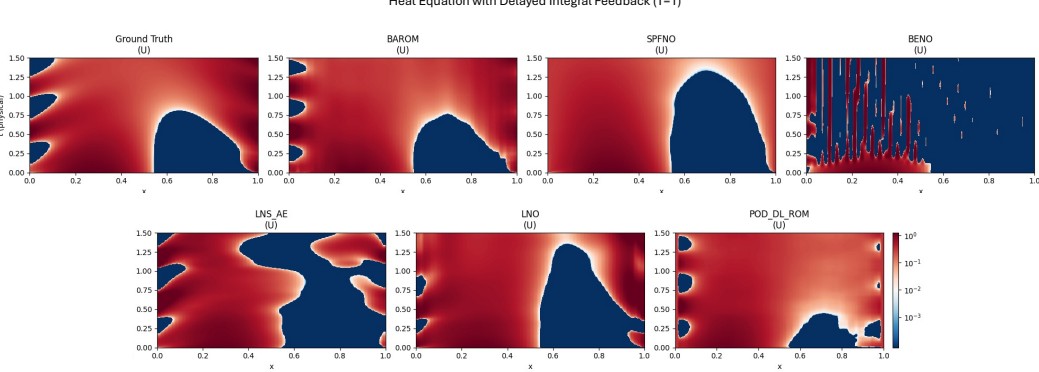

Figure 6: Visualization of prediction Heat Equation data at $T_{\text{horizon}} = 1$.

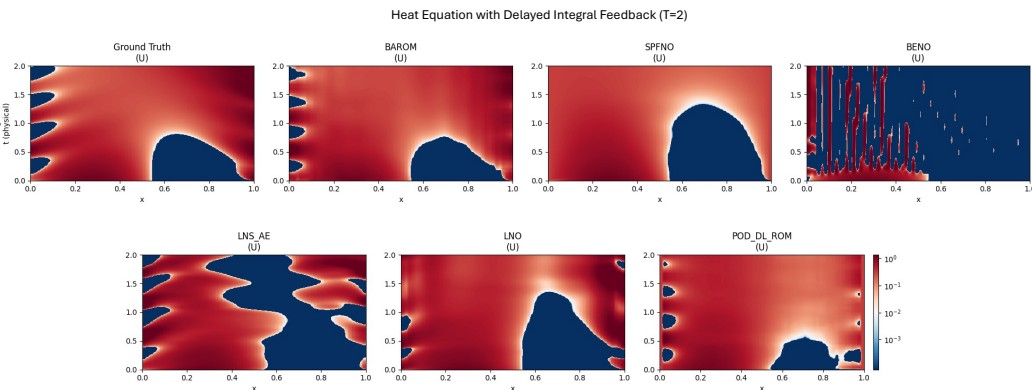

Figure 7: Visualization of prediction Heat Equation data at $T_{\text{horizon}} = 2$.

## C ADDED EXPERIMENTAL RESULTS

Table 2: Quantitative results on PDE systems without internal feedback and Boundary control. Models trained at $T = 1$, $nt = 300$, evaluated at three time-horizons.

| Model | Advection(RelErr e-1) $T=1$ | $T=1.5$ | $T=2$ | Euler(RelErr e-1) $T=1$ | $T=1.5$ | $T=2$ | Burgers(RelErre-1) $T=1$ | $T=1.5$ | $T=2$ | Darcy(RelErr e-1) $T=1$ | $T=1.5$ | $T=2$ |
|---|---|---|---|---|---|---|---|---|---|---|---|---|
| BAROM_ImpBC | **2.439** | 5.310 | 9.838 | 5.452 | 6.172 | 7.132 | 3.032 | **3.165** | **3.271** | 1.479 | 1.501 | **1.524** |
| LNO | 6.142 | 9.741 | 14.07 | 9.128 | 12.37 | 16.340 | 4.054 | 5.578 | 72.80 | 11.20 | 17.85 | 27.70 |
| POD-DL | 8.9001 | 11.68 | 14.73 | 5.8554 | 6.0410 | 6.3717 | 6.8611 | 7.0714 | 7.4849 | 2.4195 | 2.7194 | 3.5453 |
| BENO | 11.21 | 13.37 | 16.31 | 12.92 | 16.32 | 18.73 | 12.93 | 16.32 | 18.73 | 13.06 | 15.29 | 16.79 |
| SPFNO | 9.1127 | 38.389 | 104.7 | inf | inf | inf | 1.404e+11 | inf | inf | inf | inf | inf |
| LNS-AE | 7.008 | 8.063 | 9.927 | 1.337 | 1.318 | 13.06 | **2.292** | 3.937 | 5.415 | 1.714 | 1.982 | 2.111 |
| UNISOLVER | 2.441 | **4.043** | **5.021** | 6.151 | 7.809 | 7.901 | 3.421 | 4.213 | 5.121 | 1.376 | 1.487 | **1.491** |
| GNOT | 3.218 | 5.491 | 8.318 | **4.114** | **4.592** | **4.809** | 3.324 | 3.819 | 4.158 | **1.314** | **1.404** | 1.528 |

### C.1 RESULTS FOR PDE SYSTEMS WITHOUT INTERNAL FEEDBACK

This section provides the detailed quantitative results for the four PDE systems with complex, externally prescribed boundary conditions but no internal feedback loop: 1D Advection-Reaction, 1D Isothermal Euler, 1D Viscous Burgers', and 2D Darcy flow. As discussed in the main text, BAROM demonstrates robust and stable performance, particularly in long-term extrapolation where other methods may fail. The models were trained on data up to $T = 1$ ($n_t = 300$) and evaluated at three time horizons. The results are presented in Table 2.

### C.2 ABLATION STUDIES

To assess the contribution of BAROM's key components and its sensitivity to hyperparameters, we conducted a series of ablation studies on the Reaction-Diffusion Neumann Feedback (RDFNF) dataset. The results, summarized in Figure 8, validate our core architectural design choices.

**Efficacy of the Learnable Lifting Network ($\mathcal{L}_{\text{lift}}$).** Figure 8a shows a substantial improvement in prediction accuracy when using the learnable $\mathcal{L}_{\text{lift}}$ compared to a fixed, non-learnable lifting function. The Relative $L_2$ Error is markedly lower across all time horizons, underscoring the critical role of an adaptive mechanism for representing the boundary field $U_B$.

**Influence of Basis Function ($\Phi$) Initialization.** Counterintuitively, Figure 8b shows that initializing the basis functions $\Phi$ randomly leads to slightly better performance than a physics-informed POD initialization. This result highlights the power of our end-to-end training framework, which is robust enough to discover a more effective basis for predicting *dynamics* from a random start than the one prescribed by POD, which is only optimal for representing static energy in the snapshots.

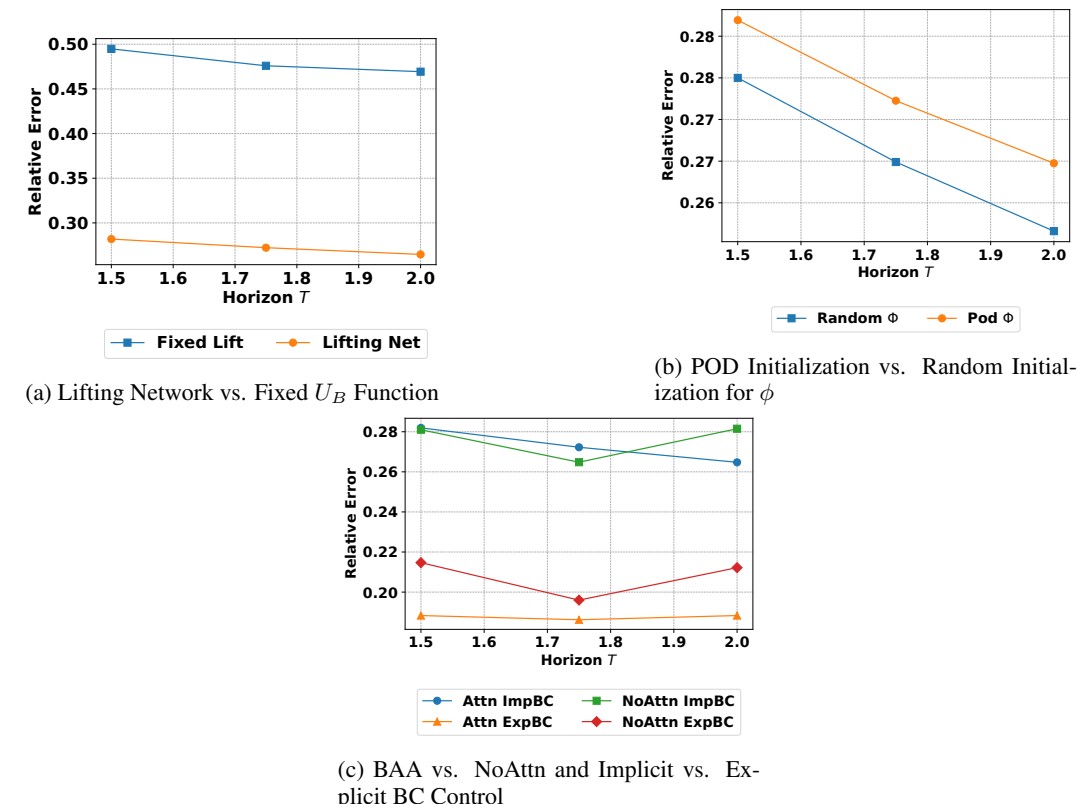

(a) Lifting Network vs. Fixed $U_B$ Function

(b) POD Initialization vs. Random Initialization for $\phi$

(c) BAA vs. NoAttn and Implicit vs. Explicit BC Control

Figure 8: Four Ablation Study *Relative Error* Vs. $T$.

**Impact of Latent Dynamics Model and BC Handling.** We compared the Boundary-Aware Attention (BAA) model with a non-attention baseline, analyzing both implicit and explicit handling of boundary parameters (Figure 8c). The results clearly show that explicitly providing boundary information to the latent dynamics model is crucial, as it significantly reduces error for both architectures. The full BAA model with explicit BC input (BAROM_ExpBC) achieved the lowest overall error, validating our final model design.

### C.3 DETAILED DISCUSSION ON EXPERIMENTAL SETUP AND FAIRNESS

To ensure a rigorous and fair evaluation, we clarify the input information provided to each model. For all baseline models (SPFNO, BENO, etc.), we adopted the standard auto-regressive approach for time-dependent problems. The model predicts the next state $\hat{U}(x, t_{k+1})$ using only information available at the current time step: the current state $U(x, t_k)$ and the current boundary parameters $P_{BC}(t_k)$.

In contrast, our primary model, **BAROM_ExpBC**, leverages boundary parameters from the *next* time step, $P_{BC}(t_{k+1})$, to evolve the latent state. This is not an ad-hoc choice for an unfair advantage, but a **core architectural feature and a central contribution of our work**, directly motivated by the established theory of the Reduced Basis Method (RBM) for non-homogeneous BCs (Abbasi et al., 2020). The classical RBM evolution equation for the latent state $\mathbf{a}^{n+1}$ explicitly requires the next-step boundary field $U_B^{n+1}$:

$$\mathbf{a}^{n+1} = \hat{A}_a \mathbf{a}^n + \hat{B}\hat{w}^n + \hat{A}_{BC} U_B^n - \mathbf{\Phi}^T U_B^{n+1} \tag{27}$$

The final term, $-\mathbf{\Phi}^T U_B^{n+1}$, is theoretically crucial for maintaining physical consistency and stability with time-varying BCs. BAROM is, to our knowledge, the first framework to implement this principle in a fully data-driven, non-intrusive manner.

For our most challenging test cases involving **internal-boundary feedback**, the required $P_{BC}(t_{k+1})$ is **not future ground-truth information**. Instead, it is dynamically computed at step $k$ using the model's *own predicted state* $\hat{U}(x, t_k)$ via our feedback module $G_{\text{fbc}}$. This computed value then serves as a known input to evolve the state to $t_{k+1}$, making the model fully auto-regressive and physically consistent without any data leakage.

Finally, to provide a direct, apples-to-apples comparison, we include our ablation model, **BAROM_ImpBC**. This variant is a standard auto-regressive model that, like the baselines, uses only the current state $U(x, t_k)$ and current boundary parameters $P_{BC}(t_k)$ to predict the next state. This allows us to isolate and quantify the performance gains derived purely from our core architectural innovations, independent of the explicit next-step BC forcing.

# D THEORETICAL JUSTIFICATION OF THE NON-LINEAR INTRINSIC EVOLUTION TERM

This appendix provides a concise derivation demonstrating *why* a scaled feed-forward network (FFN) is a suitable and theoretically grounded choice for modeling the intrinsic non-linear evolution term $\Delta\mathbf{a}_{\text{ffn}}$ used in the BAROM update rule, Eq. equation **??**. All symbols and equation numbers refer to the main text.

## D.1 FROM NON-LINEAR PDE TO NON-LINEAR LATENT SPACE ODE VIA GALERKIN PROJECTION

We begin with the general parameterized PDE as defined in Eq. equation 1:

$$\partial_t U(x, t; \mu) = \mathcal{N}(U, \nabla_x U, \dots; x, t; \mu), \qquad (x, t) \in \Omega \times \mathcal{T}, \tag{1}$$

where $\mathcal{N}$ is explicitly stated to be a **general non-linear differential operator**. The BAROM framework decomposes the solution $U = U_B + U_I$, where $U_B$ satisfies the non-homogeneous boundary conditions and the internal field $U_I$ satisfies corresponding homogeneous boundary conditions. $U_I$ is represented by the ansatz:

$$U_I(x, t; \mu) = \mathbf{\Phi}(x)\, \mathbf{a}(t; \mu), \tag{??}$$

with $\mathbf{\Phi}(x) \in \mathbb{R}^{N_x \times r}$ being the (learnable) spatial basis matrix and $\mathbf{a}(t; \mu) \in \mathbb{R}^r$ the vector of time-dependent latent coefficients.

Substituting $U = U_B + \mathbf{\Phi}\mathbf{a}$ into Eq. equation 1 yields the PDE governing the internal field $U_I$:

$$\partial_t(\mathbf{\Phi}\mathbf{a}) = \mathcal{N}(U_B + \mathbf{\Phi}\mathbf{a}, \nabla_x(U_B + \mathbf{\Phi}\mathbf{a}), \dots) - \partial_t U_B. \tag{28}$$

Assuming the basis functions $\mathbf{\Phi}(x)$ are time-independent, $\partial_t(\mathbf{\Phi}\mathbf{a}) = \mathbf{\Phi}\dot{\mathbf{a}}$. Applying a Galerkin projection onto the latent space by testing against the basis functions $\mathbf{\Phi}(x)$ (i.e., taking the inner product with $\mathbf{\Phi}_j$ for $j = 1, \dots, r$, or more compactly, pre-multiplying by $\mathbf{\Phi}^T$ and integrating over $\Omega$):

$$\int_\Omega \mathbf{\Phi}^T(\mathbf{\Phi}\dot{\mathbf{a}})\, dx = \int_\Omega \mathbf{\Phi}^T\left(\mathcal{N}(U_B + \mathbf{\Phi}\mathbf{a}, \dots) - \partial_t U_B\right)\, dx. \tag{29}$$

If we define a mass matrix $\mathbf{M} = \int_\Omega \mathbf{\Phi}^T \mathbf{\Phi}\, dx$ (which is the identity matrix $\mathbf{I}$ if $\mathbf{\Phi}$ is orthonormal with respect to the $L_2$ inner product), this simplifies to a system of ordinary differential equations (ODEs) for $\mathbf{a}(t)$:

$$\mathbf{M}\dot{\mathbf{a}}(t) = \mathbf{F}_\mathcal{N}(\mathbf{a}(t), U_B(t), \partial_t U_B(t), \mu, t), \tag{30}$$

where $\mathbf{F}_\mathcal{N}(\cdot) = \int_\Omega \mathbf{\Phi}^T\left(\mathcal{N}(U_B + \mathbf{\Phi}\mathbf{a}, \dots) - \partial_t U_B\right)\, dx$.

Crucially, if the original operator $\mathcal{N}$ is **non-linear** with respect to $U$ or its derivatives, then the term $\mathcal{N}(U_B + \mathbf{\Phi}\mathbf{a}, \dots)$ will generally contain terms that are non-linear functions of $\mathbf{\Phi}\mathbf{a}$ (e.g., $(\mathbf{\Phi}\mathbf{a})^2$, $(\mathbf{\Phi}\mathbf{a})\nabla_x(\mathbf{\Phi}\mathbf{a})$, etc.). Consequently, the projected right-hand side, $\mathbf{F}_\mathcal{N}$, will be a **non-linear function of the latent coefficients** $\mathbf{a}(t)$.

For the purpose of modeling the *intrinsic* dynamics related to $\mathbf{a}(t)$ itself, distinct from the direct forcing by boundary terms (which BAROM handles via $\Delta\mathbf{a}_{\text{bc}}$), we can represent the ODE system as:

$$\dot{\mathbf{a}}(t) = \mathbf{f}(\mathbf{a}(t); \mu, \text{boundary influences}), \tag{31}$$

where $\mathbf{f}$ (conceptually $\mathbf{M}^{-1}\mathbf{F}_{\mathcal{N}}$) encapsulates the non-linear dependencies on $\mathbf{a}(t)$ arising from $\mathcal{N}$, along with influences from $\mu$ and boundary-related terms. The $\Delta\mathbf{a}_{\text{ffn}}$ term aims to capture the non-linear self-interaction part of $\mathbf{f}$ that primarily depends on $\mathbf{a}(t)$.

## D.2 TIME DISCRETISATION OF THE NON-LINEAR LATENT ODE

Applying a first-order explicit Euler time integration scheme with step size $\Delta t > 0$ to Eq. equation 31 (focusing on the dependence on $\mathbf{a}(t_k)$ for the intrinsic part) yields:

$$\mathbf{a}(t_{k+1}) \;=\; \mathbf{a}(t_k) + \underbrace{\Delta t\,\mathbf{f}_{\text{intrinsic}}\big(\mathbf{a}(t_k);\mu\big)}_{\Delta\mathbf{a}_\star} + \text{other terms} + \mathcal{O}(\Delta t^2). \tag{32}$$

Here, $\mathbf{f}_{\text{intrinsic}}\big(\mathbf{a}(t_k);\mu\big)$ represents the component of $\mathbf{f}$ capturing the non-linear evolution due to $\mathbf{a}(t_k)$ itself. Learning an accurate approximation of the increment $\Delta\mathbf{a}_\star$ is essential for advancing the solution.

## D.3 UNIVERSAL APPROXIMATION OF THE NON-LINEAR INCREMENT BY AN FFN

The non-linear nature of $\mathbf{f}_{\text{intrinsic}}$ necessitates a non-linear function approximator. A Feed-Forward Network (FFN) is a suitable choice due to its universal approximation capabilities.

**FFN Approximation of the Latent Increment** Let $K \subset \mathbb{R}^r$ be a compact set containing the trajectory of latent coefficients $\mathbf{a}(t_k)$ encountered during training. If $\mathbf{f}_{\text{intrinsic}}(\mathbf{a};\mu)$ is continuous on $K$, then for every $\varepsilon > 0$, there exist FFN parameters $\theta = \{W_2, W_1, b_2, b_1\}$ and a scalar $\alpha > 0$ such that

$$\sup_{\mathbf{a}\in K}\big\|\alpha\,g_\theta(\mathbf{a}) - \Delta t\,\mathbf{f}_{\text{intrinsic}}(\mathbf{a};\mu)\big\|_2 \;<\; \varepsilon, \tag{33}$$

where $g_\theta(\mathbf{a}) = W_2\,\sigma(W_1\mathbf{a} + b_1) + b_2$ is a standard two-layer FFN with a suitable non-linear activation function $\sigma$ (e.g., GELU).

*Proof.* The Universal Approximation Theorem Hornik (1991) states that a two-layer FFN with a sufficient number of hidden units and a non-polynomial activation function can approximate any continuous function $\mathbf{f}_{\text{intrinsic}}(\mathbf{a};\mu)$ on a compact domain $K$ to arbitrary accuracy. Thus, there exists an FFN, let's call it $h_{\theta'}(\mathbf{a})$, such that $\sup_{\mathbf{a}\in K}\|\mathbf{f}_{\text{intrinsic}}(\mathbf{a};\mu) - h_{\theta'}(\mathbf{a})\|_2 < \varepsilon' = \varepsilon/\Delta t$. By defining $g_\theta(\mathbf{a}) = h_{\theta'}(\mathbf{a})$ (i.e., choosing appropriate FFN parameters $\theta$ for $g_\theta$) and setting the learnable scale $\alpha = \Delta t$, we achieve $\sup_{\mathbf{a}\in K}\|\alpha\,g_\theta(\mathbf{a}) - \Delta t\,\mathbf{f}_{\text{intrinsic}}(\mathbf{a};\mu)\|_2 = \Delta t\sup_{\mathbf{a}\in K}\|g_\theta(\mathbf{a}) - \mathbf{f}_{\text{intrinsic}}(\mathbf{a};\mu)\|_2 < \Delta t \cdot (\varepsilon/\Delta t) = \varepsilon$. □

## D.4 CONNECTION TO THE BAROM UPDATE RULE

The BAROM update rule, Eq. equation 12, includes the term:

$$\Delta\mathbf{a}_{\text{ffn}} \;=\; \alpha\,g_\theta\big(\mathbf{a}(t_k)\big). \tag{34}$$

As established by the theorem above, this FFN branch, with appropriately learned parameters $(\alpha, \theta)$, is provably expressive enough to approximate the non-linear increment $\Delta\mathbf{a}_\star = \Delta t\,\mathbf{f}_{\text{intrinsic}}(\mathbf{a}(t_k);\mu)$ to any prescribed accuracy $\varepsilon$.

This rigorous justification underscores the necessity and capability of the $\Delta\mathbf{a}_{\text{ffn}}$ component:

1. **Necessity**: The non-linearity of the original PDE operator $\mathcal{N}$ is generally preserved through Galerkin projection, resulting in a non-linear ODE system (Eq. equation 31) for the latent coefficients $\mathbf{a}(t)$. A linear model for this intrinsic evolution would generally be insufficient.

2. **Capability**: The FFN, as a universal approximator, can capture these inherent non-linear relationships $\mathbf{f}_{\text{intrinsic}}(\mathbf{a}(t_k))$. The learnable scale $\alpha$ further allows the model to align the FFN's output magnitude with the chosen time step $\Delta t$ and to balance its contribution against the other dynamic components ($\Delta\mathbf{a}_{\text{attn}}$ and $\Delta\mathbf{a}_{\text{bc}}$).

**Implication.** The inclusion of the $\Delta\mathbf{a}_{\text{ffn}}$ term, designed to model these projected non-linearities, is therefore crucial for the BAROM framework's ability to faithfully learn the latent space dynamics of PDE systems governed by general non-linear operators. Within the training regime, the overall latent update (Eq. equation 12) can thus approximate the true latent evolution with high fidelity, validating the architectural design choices presented in the main text.

## E IMPLEMENTATION DETAILS.

Our model, BAROM_explictBC version, integrates attention mechanisms with learned boundary condition (BC) representations for PDE solutions involving control inputs. It consists of a universal lifting network, explicitly processing BC and control inputs through separate multilayer perceptrons, and projects these into a latent state using POD-based orthogonal bases or learned initialization. The core attention block computes updates for latent coefficients via multi-head attention modules enhanced with a residual feedforward network, followed by boundary-driven updates processed through dedicated BC feature extractors. Training employs an AdamW optimizerYao et al. (2021) with a learning rate of $1 \times 10^{-4}$, gradient clipping (max norm=1.0), and a weighted composite loss combining reconstruction MSE, basis orthogonality regularization, and boundary penalties. The model was trained for 150 epochs on an NVIDIA GPU A40, leveraging CUDA acceleration and PyTorch framework. The other version model, BAROM_implicitBC version, integrates attention mechanisms with implicitly learned boundary condition (BC) representations for PDE solutions involving control inputs. It employs a universal lifting network that implicitly fuses BC and control inputs into latent states via a unified multilayer perceptron (MLP), which are then projected into latent coefficient spaces using POD-based orthogonal bases or learned initializations. The core attention module computes latent coefficient updates using multi-head attention blocks, further enhanced by residual feedforward networks to capture intricate state dynamics implicitly influenced by boundary conditions. Training is carried out using the AdamW optimizer with a learning rate of $1 \times 10^{-4}$, gradient clipping at a maximum norm of 1.0, and a composite loss function that includes reconstruction mean squared error (MSE), orthogonality regularization of the POD-based basis functions, and implicit boundary constraint penalties. The BAROM_implicitBC model was trained over 200 epoch and setting early stop strategy using an NVIDIA GPU A40, taking advantage of CUDA acceleration within the PyTorch framework.

## F SENSITIVITY ANALYSIS OF LOSS WEIGHTS

This section addresses the selection of the regularization weights, $\lambda_{\text{orth}}$ and $\lambda_{\text{bc}}$, used in the composite loss function (Eq. equation 17). Balancing these weights is crucial as it controls the trade-off between fitting the training data (via $\mathcal{L}_{\text{recon}}$) and satisfying the physical constraints on the basis functions (via $\mathcal{L}_{\text{orth}}$ and $\mathcal{L}_{\text{bc\_pen}}$).

To investigate the impact of these hyperparameters, we conducted a sensitivity analysis on the **Heat Equation with Non-linear Gain Feedback** dataset. We evaluated the model's performance by measuring the Relative $L_2$ Error (RelErr) at the end of the training horizon ($T = 1.5$) and in an extrapolation regime ($T = 2.0$). The results are presented in Table 3.

Table 3: Performance on the Heat NF dataset under different loss weights. Metrics are Relative $L_2$ Error (e-1). The configuration used in our main experiments is highlighted in **bold**.

| $\lambda_{\text{bc\_pen}}$ | T=1.5 (Interpolation) | | | T=2.0 (Extrapolation) | | |
| --- | --- | --- | --- | --- | --- | --- |
| | $\lambda_{\text{orth}}$=0.01 | $\lambda_{\text{orth}}$=0.001 | $\lambda_{\text{orth}}$=0.0001 | $\lambda_{\text{orth}}$=0.01 | $\lambda_{\text{orth}}$=0.001 | $\lambda_{\text{orth}}$=0.0001 |
| 0.1 | 2.488 | 2.415 | 2.532 | 2.653 | 2.591 | 2.712 |
| 0.01 | 2.315 | **2.280** | 2.355 | 2.495 | **2.440** | 2.545 |
| 0.001 | 2.571 | 2.498 | 2.610 | 2.784 | 2.699 | 2.833 |

**Discussion.** The analysis reveals several key trends:

- **Sensitivity to Weights:** The results confirm that the model's predictive accuracy is sensitive to the choice of these hyperparameters.

- **Impact of High Weights:** When regularization weights are set relatively high (e.g., $\lambda_{\text{bc\_pen}} = 0.1$), performance slightly degrades. We hypothesize that this imposes an overly strong constraint on the basis functions, potentially hindering their flexibility to represent the internal dynamics optimally.

- **Impact of Low Weights:** Conversely, when the weights are too low (e.g., $\lambda_{\text{bc\_pen}} = 0.001$), the regularization terms may contribute insufficiently to the total loss, which can also lead to poorer performance, possibly by not adequately enforcing the desired physical properties (orthogonality and homogeneity) on the basis functions.

In conclusion, this analysis demonstrates that selecting these weights involves a trade-off between the model's data-fitting capacity and the strength of the imposed physical priors. The parameter set used in our main experiments ($\lambda_{\text{orth}} = 0.001, \lambda_{\text{bc\_pen}} = 0.01$) represents a balanced choice that yielded the best performance in this validation study.

# G   DETAILS OF BASELINES

To ensure fair benchmarking, all baseline models were carefully adapted and tuned specifically for our PDE datasets and prediction tasks.

**Semiperiodic Fourier Neural Operator (SPFNO)**: SPFNO extends FNO by embedding explicit handling of Dirichlet and Neumann boundary conditions via spectral projection filters. Our implementation incorporated customized spectral projection operators, tailored to our datasets' boundary conditions, thus enabling more accurate boundary condition enforcement during training and inference.

**Boundary-Embedded Neural Operator (BENO)**: BENO integrates explicit boundary embeddings into neural operators, enhancing boundary-aware PDE predictions. In our setup, boundary conditions and control inputs were explicitly processed and embedded into node and edge features within a tailored graph neural network and transformer architecture, adjusted for dimensional consistency with our specific PDE state variables and controls.

**Latent Neural Operator (LNO)**: LNO utilizes a latent-space transformer architecture to learn mappings between PDE states in latent coordinate spaces, significantly reducing computational overhead. Our implementation carefully selected latent embedding dimensions and the number of latent transformer blocks, optimizing performance by fine-tuning the latent space resolution to match the complexity of our dataset scenarios.

**POD-Deep Learning Reduced Order Model (POD-DL-ROM)**: This method combines Proper Orthogonal Decomposition (POD) with deep neural networks to create efficient reduced-order representations of PDE solutions. For each PDE dataset, POD basis functions were explicitly computed and cached, followed by training specialized encoders, decoders, and deep feedforward networks, ensuring an optimal trade-off between model compactness and accuracy.

**Latent Neural Solver Autoencoder (LNS-AE)**: LNS-AE leverages an autoencoder combined with a latent neural operator to achieve efficient PDE predictions in compressed latent spaces. We carefully trained the autoencoder and latent stepper separately, optimizing autoencoder architecture parameters such as downsampling levels and latent channels specifically for capturing salient dynamics in our PDE datasets, thus ensuring the LNS-AE accurately reconstructs detailed PDE solution trajectories.

**GNOT (General Neural Operator Transformer):** GNOT is a transformer-based neural operator that uses a cross-attention, self-attention, and geometric gating structure to learn mappings between function spaces. For our time-stepping task, we adapt the GNOT architecture to predict the next state $U(x, t_{k+1})$ given the current state $U(x, t_k)$ and the boundary parameters $P_{BC}(t_k)$. The model encodes the spatial coordinates, the current state field, and the boundary parameters into a common embedding space. A series of GNOT attention blocks then processes these embeddings to produce the final prediction.

**Unisolver:**   Unisolver is a PDE-conditional transformer architecture designed to solve various PDEs by conditioning on the specific equation parameters. We adapt its core mechanism for our

time-stepping task. The model first embeds the input field $U(x, t_k)$ using a patch-based tokenizer. The boundary parameters $P_{BC}(t_k)$ are encoded into a separate conditional vector. This vector is then used to modulate the transformer blocks via adaptive layer normalization and gating, allowing the model to condition its prediction of $U(x, t_{k+1})$ on the specific boundary inputs for that time step.

## H    DIFFERENCES WITH OTHER MODELS

We compare our model, BAROM, with several representative baselines including SPFNO, BENO, LNO, LNS-AE, and POD-DL-ROM, etc. across the following dimensions, focusing on architectural distinctions beyond the setup configurations detailed in Appendix X (DETAILS OF BASELINES).

**Explicit vs. Implicit Boundary Condition (BC) Imposition**: BAROM employs an explicit ansatz decomposition ($\hat{U} = U_B + U_I$), where a learnable lifting network dynamically generates the boundary-conforming field $U_B$ from BC parameters $P_{BC}(t)$. The internal field $U_I$ is then designed for corresponding homogeneous BCs. In contrast, FNO typically handles BCs implicitly by including BC information in the input fed to its Fourier layers. SPFNO uses specific spectral transforms (sine/cosine) and projection filters for certain BC types but lacks a learnable, adaptive lifting component for $U_B$. BENO, designed for elliptic PDEs, uses GNNs and a Transformer to embed boundary geometry and values, a different paradigm from BAROM's field decomposition. LNS-AE encodes the entire solution field (including boundaries) into its latent space, with the latent stepper then using BC parameters, rather than BAROM's explicit $U_B$ separation prior to latent modeling of $U_I$. LNO integrates BC information into its branch projector before transforming to a latent space via Physics-Cross-Attention, but does not explicitly separate $U_B$. POD-DL-ROM typically applies POD to the entire solution field (possibly after a fixed lifting), with BCs influencing the DFNN that evolves the global modal coefficients, differing from BAROM's learnable lifting for $U_B$ and separate latent modeling of $U_I$.

**Latent Space Representation and Dynamics Modeling**: BAROM learns the dynamics of coefficients $\mathbf{a}(t)$ for POD-initialized, learnable basis functions $\mathbf{\Phi}(x)$ that represent $U_I$. Its attention-based dynamics model $\mathcal{F}_{\text{latent}}$ is explicitly conditioned on $P_{BC}(t)$ to ensure boundary-awareness in the evolution of $\mathbf{a}(t)$. This differs from FNO/SPFNO which operate in frequency/spectral domains. LNO learns operator dynamics between M latent points derived via PhCA, which are distinct from BAROM's modal coefficients. LNS-AE evolves a latent representation of the full field $U(x, t)$ using a stepper network. POD-DL-ROM also evolves modal coefficients of the full field, but typically with a standard DFNN, lacking BAROM's attention mechanism and the direct integration of $P_{BC}(t)$ into the latent coefficient update rule for $U_I$. BENO's GNN-based message passing and latent boundary geometry vector serve a different purpose, primarily for elliptic problems.

**Handling of Coupled and Controlled BCs**: BAROM is architecturally designed to address complex BCs, including those with internal-boundary coupling or external controls, through its learnable lifting network and the explicit conditioning of the latent dynamics on $P_{BC}(t)$. This allows $\mathcal{L}_{\text{lift}}$ to adaptively form $U_B$ and $\mathcal{F}_{\text{latent}}$ to adjust $U_I$'s evolution in response to dynamic boundary changes, including feedback. While other models like LNS-AE, LNO, and POD-DL-ROM can incorporate BC/control signals into their dynamics (e.g., as input to the latent stepper or DFNN), BAROM's structural decomposition and dedicated learnable components for $U_B$ and $U_I$ offer a more direct and potentially more robust mechanism for systems where BCs are intricately coupled with the internal state or driven by external inputs. Models like FNO and SPFNO may require more specialized adaptations to achieve similar robustness for such coupled scenarios. BENO's focus is on complex boundary geometries in elliptic PDEs rather than time-evolving controlled BCs in dynamic systems.

In particular, only BAROM synergistically combines a learnable lifting network for explicit and adaptive generation of the non-homogeneous boundary component ($U_B$) with an attention-based evolution of latent coefficients for the internal solution component ($U_I$), where these coefficients are directly modulated by real-time boundary parameters.

In summary, BAROM demonstrates clear architectural distinctions, particularly in its explicit, learnable boundary treatment and conditioned latent dynamics, making it a versatile and effective tool for time-dependent PDEs with complex, actively controlled, or feedback-dependent boundary conditions.

## I   EFFICIENCY COMPARISON

This section succinctly compares BAROM's computational efficiency with baseline models, contextualized by the predictive accuracy reported in Section 4. All metrics were recorded on identical hardware. Tables 4 and 5 present these metrics for the **HEAT_NONLINEAR_FEEDBACK_GAIN** and **CONVDIFF** datasets, respectively.

Table 4: Efficiency metrics on the HEAT_NONLINEAR_FEEDBACK_GAIN dataset. Inference time and GPU memory are averaged per sample.

| Model | Parameters | Avg. Inference Time (ms) | Peak GPU Memory (MB) |
|---|---|---|---|
| BAROM | 983,843 | 306.87 | 33.06 |
| SPFNO | 287,553 | 484.61 | 32.35 |
| BENO | 2,674,305 | 1069.63 | 32.46 |
| LNO | 179,841 | 478.20 | 32.48 |
| LNS-AE | 1,310,353 | 337.94 | 33.42 |
| POD-DL-ROM | 331,537 | 154.95 | 32.34 |

Table 5: Efficiency metrics on the CONVDIFF dataset. Inference time and GPU memory are averaged per sample.

| Model | Parameters | Avg. Inference Time (ms) | Peak GPU Memory (MB) |
|---|---|---|---|
| BAROM | 983,907 | 302.48 | 33.07 |
| SPFNO | 287,617 | 483.59 | 32.36 |
| BENO | 2,676,417 | 1069.67 | 32.48 |
| LNO | 179,905 | 478.07 | 32.49 |
| LNS-AE | 1,310,417 | 335.27 | 33.43 |
| POD-DL-ROM | 331,793 | 154.80 | 32.35 |

BAROM's parameter count (approx. 0.98M) is moderate, positioned between more lightweight models (e.g., LNO at approx. 0.18M, POD-DL-ROM at approx. 0.33M) and larger architectures (e.g., BENO at approx. 2.67M, LNS-AE at approx. 1.31M). This reflects its inclusion of specialized components for robust boundary handling.

In terms of inference speed, BAROM (approx. 300-307 ms/sample) is notably faster than several baselines such as SPFNO, BENO, and LNO, and is comparable to LNS-AE. While POD-DL-ROM shows the fastest inference, BAROM's efficiency is compelling given its significantly higher accuracy on the complex feedback-controlled PDE systems analyzed in Section **??**. The peak GPU memory usage during inference (around 32-33 MB) is similar for all evaluated models, indicating BAROM does not impose an undue memory burden.

In summary, BAROM demonstrates a strong balance between computational efficiency and its state-of-the-art predictive accuracy for challenging PDE systems with complex boundary conditions, making it a practical and effective framework.

## J   LIMITATIONS AND FUTURE WORK

While BAROM demonstrates strong performance and significant advantages in handling complex boundary conditions, we acknowledge several avenues for future research. The current work focuses on a range of challenging one- and two-dimensional systems. Extending the framework to higher-dimensional problems and more complex geometries is a natural next step. Consistent with other large-scale deep learning models, the initial training phase requires sufficient high-fidelity data and computational resources, and exploring more sample-efficient training paradigms remains an interesting direction. Finally, while our RBM-inspired design enhances physical interpretability over monolithic models, applying advanced techniques to further probe the learned neural operators presents a promising area for future investigation.

## K    HYPER-PARAMETER LIST

This section details the primary hyper-parameters used for BAROM and the baseline models. These settings were generally consistent across these three datasets for each respective model, derived from the defaults in their corresponding execution scripts. As for task 1, Common training data characteristics for these datasets include an input sequence length corresponding to $T_{train} = 1$ (which is 300 time steps, from a total of 600 steps for $T = 2.0$). As for task 2, Common training data characteristics for these datasets include an input sequence length corresponding to $T_{train} = 1.5$ (which is 225 time steps, from a total of 300 steps for $T = 2.0$).

### K.1    BAROM

Table 6: Hyper-parameters for BAROM.

| Parameter Type | Value |
| --- | --- |
| *Architectural Parameters* | |
| Basis Dimension (Coefficients) | 32 |
| Attention: Model Dimension | 512 |
| Attention: Number of Heads | 4 |
| Boundary Feature Processor(BAROM_ExpBC Only): Output Dimension | 64 |
| Boundary Feature Processor(BAROM_ExpBC Only): Hidden Dimension | 128 |
| Lifting Network: State Branch Hidden Dimension | 32 |
| Lifting Network: Control MLP Hidden Dimensions | [32, 128] |
| Lifting Network: Fusion MLP Hidden Dimensions | [256, 512, 256] |
| Intrinsic Update FFN: Hidden Dimension | 256 |
| Intrinsic Update FFN: Number of Layers | 3 |
| *Training Parameters* | |
| Epochs | 150 |
| Learning Rate | 5e-4 |
| Batch Size | 32 |
| Optimizer | AdamW |
| Weight Decay | 1e-5 |
| Scheduler | ReduceLROnPlateau |
|     Factor | 0.5 |
|     Patience | 10 |
| Loss Weight: Basis Orthogonality | 0.001 |
| Loss Weight: Basis BC Penalty | 0.01 |
| Gradient Clip Norm | 1.0 |

## K.2 SPFNO (Semi-periodic Fourier Neural Operator)

Table 7: Hyper-parameters for SPFNO.

| Parameter Type | Value |
|---|---|
| *Architectural Parameters* | |
| Number of Fourier Modes | 16 |
| SPFNO Layer Width (Channels) | 64 |
| Number of SPFNO Layers | 4 |
| Spectral Transform Type | 'dirichlet' |
| Use Projection Filter | True |
| *Training Parameters* | |
| Epochs | 150 |
| Learning Rate | 1e-3 |
| Batch Size | 32 |
| Optimizer | AdamW |
| Weight Decay | 1e-4 |
| Scheduler | ReduceLROnPlateau |
| Factor | 0.5 |
| Patience | 10 |
| Gradient Clip Norm | 1.0 |

### K.3 BENO (BOUNDARY-EMBEDDED NEURAL OPERATOR)

Table 8: Hyper-parameters for BENO.

| Parameter Type | Value |
| --- | --- |
| *Architectural Parameters* | |
| Embedding Dimension | 64 |
| GNN Hidden Dimension | 64 |
| Number of GNN Layers | 4 |
| Boundary Embedder: Transformer Layers | 2 |
| Boundary Embedder: Attention Heads | 4 |
| *Training Parameters* | |
| Epochs | 150 |
| Learning Rate | 5e-4 |
| Batch Size | 32 |
| Optimizer | AdamW |
| Weight Decay | 1e-4 |
| Scheduler | ReduceLROnPlateau |
|   Factor | 0.5 |
|   Patience | 10 |
| Gradient Clip Norm | 1.0 |

### K.4 POD-DL-ROM

Table 9: Hyper-parameters for POD-DL-ROM.

| Parameter Type | Value |
| --- | --- |
| *Architectural Parameters* | |
| POD Basis Dimension | 64 |
| CAE Latent Dimension | 8 |
| Encoder: Convolutional Channels | [16, 32, 64] |
| Encoder: Fully Connected Layers | [128] |
| DFNN for Dynamics: Hidden Layers | [256, 512, 256] |
| *Training Parameters* | |
| Epochs | 150 |
| Learning Rate | 1e-4 |
| Batch Size | 32 |
| Loss Weighting (Rec. vs. Intrinsic) | 0.5 |
| Optimizer | AdamW |
| Weight Decay | 1e-4 |
| Scheduler | ReduceLROnPlateau |
|   Factor | 0.5 |
|   Patience | 10 |
| Gradient Clip Norm | 1.0 |

## K.5 LNO (LATENT NEURAL OPERATOR)

Table 10: Hyper-parameters for LNO.

| Parameter Type | Value |
|---|---|
| *Architectural Parameters* | |
| Number of Latent Points | 64 |
| Embedding Dimension | 64 |
| Number of Transformer Blocks | 3 |
| Transformer Heads | 4 |
| Transformer Feedforward Dimension | 128 |
| Projector Hidden Dimensions | [64, 128] |
| Final MLP Hidden Dimensions | [128, 64] |
| Input Coordinate Dimension | 2 |
| *Training Parameters* | |
| Epochs | 150 |
| Learning Rate | 3e-4 |
| Batch Size | 32 |
| Optimizer | AdamW |
| Weight Decay | 1e-5 |
| Scheduler | ReduceLROnPlateau |
| Factor | 0.5 |
| Patience | 10 |
| Gradient Clip Norm | 1.0 |

## K.6 LNS-AE (LATENT NEURAL SOLVER WITH AUTOENCODER)

Table 11: Hyper-parameters for LNS-AE.

| Component | Parameter Type | Value |
|---|---|---|
| *Autoencoder* | | |
| | Initial Width | 64 |
| | Downsampling Blocks | 3 |
| | Latent Channels | 16 |
| | Final Latent Spatial Dimension | 8 (for input nx=64) |
| | Epochs | 100 |
| | Learning Rate | 3e-4 |
| | Optimizer | AdamW |
| | Scheduler | ReduceLROnPlateau |
| *Latent Stepper* | | |
| | Branch Hidden Dimensions | [128, 128] |
| | Trunk Hidden Dimensions | [64, 64] |
| | Combined Output Projection Dim. | 128 |
| | Epochs | 150 |
| | Learning Rate | 1e-4 |
| | Training Rollout Steps | 1 |
| | Optimizer | AdamW |
| | Scheduler | CosineAnnealingLR |
| *Common* | | |
| | Batch Size | 32 |
| | Gradient Clip Norm | 1.0 |

Table 12: Hyper-parameters for GNOT-Stepper.

| Parameter Type | Value |
|---|---|
| *Architectural Parameters* | |
| Embedding Dimension | 128 |
| Number of GNOT Layers | 4 |
| Attention: Number of Heads | 8 |
| Geometric Gating: Num Experts | 4 |
| MLP Hidden Dimensions (Encoders) | [128, 128] |
| *Training Parameters* | |
| Epochs | 150 |
| Learning Rate | 5e-4 |
| Batch Size | 16 |
| Optimizer | AdamW |
| Weight Decay | 1e-4 |
| Scheduler | ReduceLROnPlateau |
|   Factor | 0.5 |
|   Patience | 10 |
| Gradient Clip Norm | 1.0 |

Table 13: Hyper-parameters for Unisolver-Stepper.

| Parameter Type | Value |
|---|---|
| *Architectural Parameters* | |
| Embedding Dimension | 256 |
| Transformer Depth (Layers) | 8 |
| Attention: Number of Heads | 8 |
| MLP Hidden Dimension (in Block) | 512 |
| Patch Size | 8 |
| *Training Parameters* | |
| Epochs | 150 |
| Learning Rate | 5e-4 |
| Batch Size | 16 |
| Optimizer | AdamW |
| Weight Decay | 1e-4 |
| Scheduler | ReduceLROnPlateau |
|   Factor | 0.5 |
|   Patience | 10 |
| Gradient Clip Norm | 1.0 |

## L  BROADER IMPACTS

The BAROM framework, for simulating PDEs with complex boundary conditions, presents potential societal impacts.

### L.1  POTENTIAL POSITIVE IMPACTS

BAROM can accelerate scientific and engineering innovation by enabling faster, more accurate simulations in fields like sustainable energy, climate modeling, and biomedical engineering. It can improve industrial design, optimization, and real-time control, leading to more efficient processes and safer systems. Furthermore, efficient ROMs like BAROM may democratize access to advanced simulation, fostering broader research and educational benefits.

### L.2  POTENTIAL NEGATIVE IMPACTS AND CONSIDERATIONS

Advanced simulation capabilities could theoretically be misused if not governed by ethical principles (e.g., in applications with harmful environmental or security implications). As with automation, widespread adoption might shift workforce demands, necessitating retraining initiatives. Over-reliance on model predictions without understanding their limitations or biases could lead to flawed decisions; thus, user education and model validation are crucial. While aiming for efficiency, the development of such models still requires resources, potentially widening gaps if access isn't democratized.

**Ethical Stance.**  The authors intend BAROM for beneficial scientific and engineering advancements and advocate for its responsible use, encouraging ongoing dialogue on ethical AI in simulation.

## M  STATEMENT ON THE USE OF LARGE LANGUAGE MODELS (LLMS)

During the preparation of this manuscript, we utilized a Large Language Model (LLM) as a general-purpose writing assistance tool. The use of the LLM was strictly limited to improving the quality and clarity of the English prose.

Specifically, the LLM was employed for the following tasks:

- Proofreading to identify and correct typographical errors.
- Correcting grammatical mistakes and ensuring syntactical correctness.
- Rephrasing sentences to improve readability, flow, and conciseness.

