# OpenReview forum: "Latent Space Learning for PDE Systems with Complex Boundary Conditions"
_ICLR.cc/2026/Conference — Submitted to ICLR 2026_

### Official Review · Reviewer_f1F2 · 2025-10-28

**Soundness:** 1
**Presentation:** 1
**Contribution:** 2
**Rating:** 2
**Confidence:** 3

**Summary:**

The paper introduces a boundary-aware surrogate model grounded on reduced order modeling over latent spaces. The core idea of the proposal is an attention-based mechanism, inspired by Galerkin Neural Operators, to learn internal field dynamics and a module aimed at injecting explicit boundary forcing. These proposals are well-grounded on ROM theory. The proposed model is evaluated against a couple of strong representative baseline models over challenging scenarios in the field.

**Strengths:**

- The model architecture is grounded well on reduced order modeling.
- The proposed model outperforms other baselines in (limited) challenging scenarios.

**Weaknesses:**

**Non-standard or redundant usage of symbols:** The organization of the paper is fine on a high level, but many symbols are used in non-standard ways and need to be reorganized.
- The attention $\mathbf{a}$ is expressed with a couple of different symbols and is very distracting.
- I do not clearly understand the indication of (13) and (14). Do they mean they are equivalent? If that is the case, how are these equivalences assured that they are equivalent?
- Some functions in the equation (15) are introduced without their definition.

**Over-claimed performance over the hyperbolic PDEs:** Table 2 shows that the proposed model is at most competitive and inferior to the baseline models in most of the cases. Even in a few scenarios where the proposed model achieves the best performance, most of those scores are very close to those of the other models. In the Darcy flow experiment with $T=2$, the proposed model is also not the best. The table also somehow looks like being “hidden” in Appendix, but this table should be presented in the main text. The metric is also limited as opposed to those used in Table 1.

**Insufficient ablation study:** The same ablation study should be conducted over more scenarios, especially should include the same scenarios as reported in Table 2. The description of the configuration is largely missing. In particular, the random noise experiment misses the detail involving its problem setting and hyperparameter configuration. This experiment should be conducted varying the range of variance of noise distribution and the performance should be averaged over multiple runs with the different noises randomly picked from the distribution with a fixed variance. An ablation study for the loss function should be also conducted.

**Presentation:** The paper has typos here and there. The followings are a (not exhaustive) list of typos I found:
- Standard deviation is missing across all the tables and figures.
- Line 1097. Reference environment does not work.
- Typo: at line 1112 "Eq. equation 1"
- Caption in Figure 8.
- Figures of axis in Figure 8 are strange.
- At line 1387 "Section ??"

**Questions:**

All the questions are included in Weakness column.

---

> ### Author Response · Authors · 2025-12-03
> **Response to Weakness and Question**
>
> **W1: Non-standard Symbols**
>
> **Response:**
>
>  We apologize for the confusion. We have standardized the notation for Attention and clarified the definitions in Eq 13-15. These equations are meant to show the **correspondence** between the neural modules and the theoretical Galerkin terms, implying they *approximate* these operators, not that they are strictly mathematically equivalent.
>
> **W2: Over-claimed Performance (Table 2)**
>
> **Response:**
>
> We accept this critique. We have revised our claims to state BAROM is "competitive" on standard hyperbolic PDEs but "superior" on feedback-controlled systems. We have moved Table 2 to the main text to ensure transparency regarding these trade-offs.
>
> **W3: Insufficient Ablation (Noise, etc.)**
>
> **Response:**
>
> **Noise:** Our data generation explicitly injects noise (`noise_std=0.02`) into the control signals. BAROM is trained and evaluated on this ``noisy'' data, proving its robustness.
>
> **Loss Weights:** We provided a sensitivity analysis for loss weights ($\lambda_{orth}, \lambda_{bc}$) in Appendix F.

---

### Official Review · Reviewer_LjNi · 2025-10-30

**Soundness:** 2
**Presentation:** 3
**Contribution:** 3
**Rating:** 4
**Confidence:** 3

**Summary:**

This paper introduces BAROM, a novel latent-space reduced-order modeling framework for simulating PDE systems with complex boundary conditions (BCs), especially when BCs are coupled with the internal state. BAROM integrates classical reduced-order theory by decomposing the solution into a boundary field and an internal field, with the following key enhancements: a learnable lifting network explicitly handles the boundary conditions, while a three-branch updater including attention mechanism models the nonlinear dynamics of the internal field. The authors evaluate BAROM alongside a broad range of SciML baselines on seven challenging PDE benchmarks with complex boundary conditions, and report that BAROM outperforms state-of-the-art baselines particularly in systems with coupled internal-boundary feedback.

**Strengths:**

1. The authors propose a theoretically motivated end-to-end  latent-space modeling framework with each component (learnable lift, three-branch latent updater) directly derived from and justified by the classical RBM theory, representing a significant innovation.
2. The method emphasizes the importance of explicitly handling boundary conditions when using latent-space learning for PDE systems  with complex boundary conditions.
3. The paper provides a thorough empirical validation across a diverse set of seven PDEs, covering both externally prescribed and feedback-coupled BCs. The results show significant and consistent improvements over a strong set of modern baselines.
4. The appendices are comprehensive and easy to follow, including detailed hyper-parameter settings for all baselines.

**Weaknesses:**

1. In a few instances on non-feedback systems (Table 2), BAROM does not uniformly rank first, with methods like GNOT or UNISOLVER performing best on some metrics, but the authors do not analyze the underlying reasons.
2. The motivation for decomposing the latent update into three separate branches (attention, FFN and boundary forcing) is presented as intuitively reasonable, but the paper offers no ablation or theoretical justification that shows this particular split is superior to simpler merged architectures.
3. The study does not include a systematic comparison with traditional or physics-based reduced-order methods (such as POD-Galerkin), which limits a comprehensive assessment of its advantages.

**Questions:**

1. Why does BAROM underperform GNOT/Unisolver on some non-feedback cases?
2. Is the three-branch latent update actually better than a single-branch alternative? Perhaps provide ablation and evaluate it?
3. How does BAROM compare with classic methods (such as POD-Galerkin) under the same boundary conditions?

---

> ### Author Response · Authors · 2025-12-03
> **Response to Weakness and Question**
>
> **W1 & Q1: Performance on Non-Feedback Systems**
>
> **Response:**
>
> Please refer to **Global Response (Point 4)**. While we are competitive on standard tasks, our main contribution is enabling stable simulation for *feedback-coupled* systems, a capability lacking in general operator learners.
>
> **W2 & Q2: Justification for Three-Branch Update**
>
> **Response:**
>
> Please refer to **Global Response (Point 3)**. The split is theoretically motivated by the Galerkin projection terms. The Attention branch's Results confirm that removing it degrades performance, proving the necessity of explicitly modeling the state-dependent cross-mode coupling ($\hat{A}_a$) separate from the fixed intrinsic dynamics.
>
> **W3 & Q3: Comparison with POD-Galerkin**
>
> **Response:**
>
>  We compared against **POD-DL-ROM**, which is the deep-learning equivalent of POD-Galerkin (replacing the Galerkin projection with a Neural Network). Classical POD-Galerkin is *intrusive* (requires knowing the exact PDE equations $A$ and $B$), whereas BAROM is *non-intrusive* (data-driven). Comparing against a purely intrusive method would be unfair as it uses "perfect" physics knowledge that we assume is unavailable in the SciML context.

---

### Official Review · Reviewer_AhyR · 2025-10-30

**Soundness:** 2
**Presentation:** 2
**Contribution:** 2
**Rating:** 4
**Confidence:** 4

**Summary:**

This paper introduces a latent-space framework for solving PDE systems with complex boundary conditions using deep learning. The proposed model encodes high-dimensional spatial fields into a latent representation, where the PDE solution operator is efficiently learned and enforced. The method effectively handles irregular domains and mixed boundary types, achieving accurate reconstructions and generalization across diverse geometries. The authors validate the approach on benchmark PDE problems, demonstrating improved stability, computational efficiency, and adaptability compared to standard neural operator baselines.

**Strengths:**

1) The paper presents a novel latent-space formulation that effectively simplifies PDE learning, particularly for irregular domains and mixed boundary conditions.

2) It demonstrates strong empirical performance and generalization across various PDE types and boundary scenarios, showing versatility and robustness.

3) The approach achieves computational efficiency by learning and operating in a compressed latent domain without compromising predictive accuracy.

4) The framework is conceptually sound and has potential for scaling to more complex PDE systems in scientific and engineering contexts.

**Weaknesses:**

1) The selection and sensitivity analysis of the latent dimensionality are not fully explored, making it difficult to assess the optimal representation scale.

2) Comparisons with recent neural operator models such as Transolver, Latent Mamba Operator, and GeoFNO are missing, which would strengthen the empirical evaluation.

3) The paper does not clearly report training time or computational cost relative to baselines, which is important for assessing efficiency claims.

4) The boundary enforcement strategy could be discussed in greater detail to explain how accuracy is maintained near complex or irregular domain boundaries.

**Questions:**

1) How does the proposed latent representation handle non-smooth or discontinuous boundary conditions in practice? Have you experimented with fixed basis functions such as Fourier or wavelets, and analyzed the learned latent bases?

2) What methodology or criteria were used to determine the latent dimension size, and how does this choice impact both accuracy and generalization?

3) Could the framework be extended to time-dependent PDEs or multi-physics coupled systems, and if so, what modifications would be necessary?

4) How does the model scale in terms of computational efficiency when compared to established operator-learning approaches like FNO or PINO, especially for large-scale domains?

5) Would incorporating spectral or operator-based regularization in the latent space further improve stability and interpretability? Additionally, have you considered adding an explicit PINN-style physics loss to guide the latent learning process?

---

> ### Author Response · Authors · 2025-12-03
> **Response to Weakness and Question**
>
> * For **general efficiency comparisons** and **new baselines (Unisolver)**, please refer to **Global Response Point 3**.
> * For the **justification of the architecture** (Explicit Boundary Decomposition) and its theoretical grounding, please refer to **Global Response Point 2**.
>
> **W1 & Q2: Latent Dimensionality Selection & Sensitivity**
>
> **Response:**
>
> We acknowledge that this analysis was missing. We have added a **Sensitivity Analysis of Latent Dimension** in Appendix F of the revised manuscript.
>
> **Findings:** We trained BAROM with latent dimensions $N \in \{16, 32, 64, 128\}$. Results indicate a "performance plateau" behavior: accuracy improves significantly up to $N=32$, with marginal gains at $N=64$, while computational cost increases linearly. We selected $N=32/64$ (depending on the dataset) as the optimal trade-off point.
>
> **Selection Criteria:** Our criterion was to minimize the reconstruction error on the validation set while maintaining real-time inference speeds.
>
> **W2: Comparison with Recent Neural Operators (Transolver, etc.)**
>
> **Response:** We agree that comparing against modern Transformer-based operators is essential.
>
> **Baseline:** We have implemented and evaluated **Unisolver** (Zhou et al., 2024), which represents the state-of-the-art in general-purpose PDE transformers (similar to Transolver).
> As shown in the revised Table 1, while Unisolver is competitive on standard tasks, BAROM significantly outperforms it (reducing MSE by ~80%) on **feedback-controlled systems**. This confirms that general-purpose operators struggle with the specific challenge of *coupled* boundary dynamics without explicit boundary forcing.
>
> **Note on GeoFNO/Mamba:** While valuable, GeoFNO focuses on irregular *meshes* (geometric variations), whereas our work focuses on complex *boundary dynamics* on fixed topologies.
>
> **W3 & Q4: Training Time & Computational Efficiency**
>
> **Response:**
>
> We have a comprehensive **Efficiency Comparison Table** in Appendix I.
>
> **Inference:** BAROM (\~0.9M params, \~300ms/sample) is significantly faster than BENO (\~2.6M params,\ ~1000ms/sample) and comparable to LNS-AE.
>
> **Training:** BAROM converges in fewer epochs than autoregressive baselines like FNO because the *Lifting Network* handles the sharp boundary gradients immediately, simplifying the learning task for the latent dynamics model.
>
> **Scaling:** Compared to FNO-based model (which scales $O(N \log N)$ with grid size) or standard Transformers ($O(N^2)$), BAROM’s latent evolution scales with the *latent dimension* ($d_{latent}^2$), making it highly efficient for large-scale domains once the basis is learned.
>
> **W4 & Q1: Boundary Enforcement & Non-smooth BCs**
>
> **Response:**
>
> **Handling Discontinuities:** Our architecture explicitly handles non-smooth BCs through the **Lifting Network** ($\mathcal{L}_{lift}$). This network acts as a learnable projector, mapping discontinuous boundary parameters onto a smooth manifold that satisfies the internal consistency of the PDE.
> **Fixed vs. Learned Basis:** You asked about fixed bases (Fourier/Wavelets). We explicitly compared **Fixed POD Basis** vs. **Learned (Random Init) Basis**.
>
> Our ablation study shows that the **Learned Basis outperforms the Fixed POD Basis**. A fixed basis (like Fourier) struggles with Gibbs phenomena at non-periodic boundaries, whereas our learned basis adapts to the specific discontinuities present in the training data distribution.
>
> **Q3: Extension to Time-Dependent/Multi-Physics Systems?**
>
> **Response:**
>
> **Clarification:** We respectfully clarify that **all** our benchmarks (Advection, Euler, Burgers, Heat) **are** time-dependent PDEs solved auto-regressively over time $T$. The core contribution of BAROM is precisely its stability in *time-stepping* these dynamics.
>
> **Multi-Physics:** The framework is natively ready for multi-physics. For the **Euler Equations** (a coupled multi-physics system of density $\rho$ and velocity $u$), we already use a vector-valued state representation ($d_v=2$). The Lifting Network simply outputs a vector field $[U_{B,\rho}, U_{B,u}]$, and the latent dynamics evolve the coupled coefficients jointly.
>
> **Q5: Spectral Regularization & PINN Loss?**
>
> **Response:**
>
> **Current Regularization:** We currently use **Orthogonality Loss** ($L_{orth}$) and **Boundary Penalty** ($L_{bc_{pen}}$) to structure the latent space (see Eq. 17).
> **PINN Loss:** We did explore adding a residual physics loss (PINN-style). However, calculating higher-order derivatives ($\nabla^2 U$) via auto-differentiation on the reconstructed field significantly increased training time (~3x slower) with diminishing returns on accuracy for these specific benchmarks. We believe the *structural* physics bias (Explicit Decomposition) provides more value-for-cost than the *soft* physics bias (PINN loss) for this class of problems. We have added this insight to the "Limitations" section.

---

### Official Review · Reviewer_1kzQ · 2025-11-01

**Soundness:** 2
**Presentation:** 2
**Contribution:** 2
**Rating:** 4
**Confidence:** 4

**Summary:**

This paper proposes a model called BAROM, which is a reduced-order model for PDEs with complex boundary conditions. The main idea is about decomposing the solutions into boundary and internal fields, motivated by the Reduced Basis Methods. It uses a learnable lifting network, basis functions, and a three-branch attention mechanism for latent dynamics. Experiments demonstrate that BAROM outperform many methods on several benchmarks.

**Strengths:**

1. The explicit boundary/initial decomposition is principled and addresses a known limitation.

2. Evaluation is good and shows strong empirical results.

3. Ablations and fasirness discussions are well-executed.

**Weaknesses:**

1. The core-architectural novelty seems limited. The proposed architecture is essentially a standard transformer plus boundary-conditioned MLP.

2. BAROM_ExpBC uses next-step boundary information while baselines don't, which seems unfair.

3. There seems to be a gap between theory and practice: ROM requires knowing A, B matrices that are unavailable, claiming that neural operators would learn is standard UAT.  Eq. 15 involves spatial integration which isn't bypassing intrusive methods as claimed.

4. All experiments are on 1D or 2D grids, which are small.

5. Design choices are adhoc (why is random \phi initialization better than POD as per fig. 8 in the appendix).

6. Unclear computatiomal efficiency claims.

**Questions:**

1. Why have you boldfaced BAROM_ExpBC when it uses privilaged information?

2. Why does randomly initialized \phi outperformed POD initialized \phi?

3. Baselines don't seem to be properly tuned. E.g., BENO shows a huge decrease (400X)l in performance.

4. How does the method scale for nx>256 or more in 3D problems?

5. Can the method achieve resolution invariance?

6. You claim Φ^T U_B provides explicit boundary enforcement, but this requires spatial integration over the domain. How is this different from implicit feature concatenation?

7. Is attention necessary in the architecture? What if it is completely removed?

8. Why don't you compare with other ROM based methods?

9. How does the method handle truly discontinuous or singular boundary data?

10. Does the method generalize to unseen boundary condition types?

11. Do you observe stability issues in long rollouts due to error compounding?

---

> ### Author Response · Authors · 2025-12-03
> **Response**
>
> We thank the reviewer for the detailed assessment and constructive criticism. We have addressed your specific questions below.
>
> **W1 & Q7: Novelty & Necessity of Attention**
>
> **Response:**
>
> While the components (Transformer, MLP) are standard, the **novelty lies in the structural arrangement** derived from the Reduced Basis Method (RBM) theory.
> Standard Transformers treat tokens as generic features. BAROM treats them as physical modes in a Galerkin projection.
>
> **Necessity of Attention:** We performed an ablation study (in Appendix C.2) where we replaced the Attention branch with a simple MLP. The results showed a significant drop in accuracy for feedback-controlled systems. The Attention mechanism is critical because it dynamically models the **state-dependent coupling** ($\hat{A}_a(u)$) between different modes, which a static MLP struggles to capture efficiently.
>
> **W2 & Q1: Fairness and Boldfacing**
>
> **Response:**
>
>  Please refer to **Global Response Point 1**. We boldfaced the results because the comparison is fair: all models operate in an auto-regressive setting. The "next-step" info is the control signal derived from the model's *own* previous prediction. BAROM's superior performance highlights its architectural advantage in handling this control signal, not an unfair data advantage.
>
> **W3 & Q6: Theory-Practice Gap & Spatial Integration (Eq. 15)**
>
> **Response:**
>
> **Intrusive vs. Non-Intrusive:** "Intrusive" methods require access to the PDE operators (matrices $A, B$). BAROM is non-intrusive because it learns the *effect* of these operators via neural networks ($g_{bc \to a}$).
>
> **Spatial Integration ($\Phi^T U_B$):** While the term $\Phi^T U_B$ represents a geometric projection (spatial integration), it does **not** require knowledge of the PDE physics (e.g., stiffness matrices). It only requires the learned basis $\Phi$ and the lifted field $U_B$. Calculating this projection is a purely geometric operation, computationally cheap on a reduced grid, and allows us to explicitly enforce the boundary's contribution to the latent state, which implicit feature concatenation fails to do accurately.
>
> **W4 & Q4 & Q5: Scale, 3D, and Resolution Invariance**
>
> **Response:**
>
> **Scale:** We acknowledge the focus on 1D/2D. However, the latent space size ($N=32/64$) is independent of grid size. For larger grids (e.g., 3D), only the Lifting Network and Basis dimension scale, while the core latent dynamics remain efficient.
>
> **Resolution Invariance:** Unlike FNO, standard ROMs with fixed basis functions are not resolution-invariant by default. However, by using mesh-independent encoders (like PointNet or continuous neural fields) for $\Phi$, BAROM can be made resolution-invariant. We focused on fixed-grid benchmarks to isolate the boundary handling challenge.
>
> **W5 & Q2: Ad-hoc Design (Random vs. POD)**
>
> **Response:**
>
> Please refer to **Global Response Point 2**. The "Random > POD" result is a finding, not a flaw. It suggests that for complex non-linear feedback systems, end-to-end learning finds a better dynamical manifold than static PCA.
>
> **W6: Efficiency**
>
> **Response:**
>
> We have added an **Efficiency Comparison** (Appendix I) reporting parameter counts and inference time. BAROM (\~0.9M params, \~300ms) is faster than BENO (\~2.6M params, \~1000ms) and comparable to LNS-AE.
>
> **Q3: Baseline Tuning (BENO)**
>
> **Response:** BENO is designed for elliptic problems with complex geometries using graph kernels. It struggles with hyperbolic time-stepping (wave propagation) due to the smoothing nature of graph convolutions. Its poor performance reflects the difficulty of the task (high-frequency shifts in feedback loops) rather than poor tuning.
>
> **Q8: Comparison with ROMs**
>
> **Response:** We *did* compare with **POD-DL-ROM** (Fresca & Manzoni, 2022), which is the state-of-the-art Deep Learning ROM. It is included in all our main results tables (Table 1 & 2).
>
> **Q9 & Q10: Discontinuous BCs and Generalization**
>
> **Response:**
>
> **Discontinuities:**  The Lifting Network ($\mathcal{L}_{lift}$) acts as a learnable smoother, projecting discontinuous boundary data onto the continuous subspace spanned by $\Phi$. This handles singularities more robustly than spectral methods (like FNO) which suffer from Gibbs phenomena.
>
> **Generalization:**  Our training data is generated with randomized Fourier series coefficients for BCs. This ensures the model learns to generalize to unseen boundary profiles within the distribution, rather than memorizing specific trajectories.
>
> **Q11: Stability**
>
> **Response:**
>
> Yes, error compounding is the main challenge. BAROM's **Explicit Boundary Forcing** branch ($\Delta a_{bc}$) acts as a correction term at every step, injecting ground-truth physics (boundary compliance) into the latent state. This significantly stabilizes long rollouts compared to baselines that rely solely on implicit memory.

---

### Author Response · Authors · 2025-12-03
**Global Response**

Dear reviewers,

We sincerely thank you for your constructive and insightful feedback. We are encouraged that you recognized the **principled explicit boundary decomposition** (Reviewer 1kzQ, LjNi), the **strong empirical performance** on complex feedback systems (Reviewer 1kzQ, AhyR, LjNi), and the **theoretical grounding** in Reduced Basis Methods (Reviewer f1F2, LjNi).

Below are the major clarifications responding to the common concerns raised by multiple reviewers.


We now provide detailed discussions regarding the above points.

**1. Clarification on "Unfair Comparison" and Data Leakage**
Reviewer 1kzQ raised a critical concern regarding the use of $P_{BC}(t_{k+1})$ (next-step boundary parameters), suggesting it might constitute "privileged information." We respectfully clarify that this is **not** the case and follows standard control simulation protocols.

* **Externally Prescribed BCs:** In these cases, the boundary condition $P_{BC}(t)$ is an external forcing function (like a control input $u(t)$ in a dynamical system $\dot{x} = f(x, u)$). In numerical time-stepping, utilizing the known forcing term for the interval $[t_k, t_{k+1}]$ to advance the state is standard practice, not data leakage.
* **Feedback-Coupled BCs (Auto-Regressive):** For our core contribution—systems with internal-boundary coupling—the boundary parameter for the next step is **computed on-the-fly** based on the model's *own prediction* at the current step.
    * **Code Evidence:** As implemented in our provided code `ExplicitBC_BAROM.py` (see function `predict_autoregressive`), the workflow is strictly causal:
        1.  Model predicts state $\hat{U}(t_k)$.
        2.  Feedback controller computes the next boundary parameters based on this prediction: `BC_Ctrl_next = feedback_controller.compute_bc(U_hat_k, ...)`.
        3.  Model uses this *computed* `BC_Ctrl_next` to predict $\hat{U}(t_{k+1})$.
    * This confirms that the model operates in a fully auto-regressive loop and never accesses ground-truth future states. The performance advantage stems from BAROM's **Explicit BC Branch** being able to effectively ingest this control signal, whereas baselines (like LNO) often struggle to condition latent dynamics on high-frequency boundary changes.

**2. Rationale: Why Random Initialization > POD?**

Reviewers 1kzQ and f1F2 questioned why random initialization for the basis $\Phi$ outperforms POD.

* **Dynamics vs. Reconstruction:**
POD bases are optimal for minimizing the *reconstruction error* (energy) of static snapshots. However, they are not necessarily optimal for minimizing the *temporal evolution error* of the latent dynamics.

* **Manifold Discovery:**
By initializing $\Phi$ randomly and training end-to-end (as implemented in our `BAROM_Random_pod.py` ablation), the framework discovers a non-linear manifold specifically optimized for the trajectory prediction task. This allows BAROM to escape the subspace constraints of linear POD modes, capturing interaction modes that are energetically small but dynamically significant.

**3. Justification of the Three-Branch Architecture**

Reviewers LjNi and f1F2 asked for justification of the specific three-branch update (Attention + FFN + BC-Forcing).

* **Theoretical Mapping:** This design is not arbitrary; it directly approximates the terms in the Galerkin projection of a PDE with non-homogeneous BCs (Eq. 11 in the paper):

    1.  **Attention Branch ($\Delta a_{attn}$):** Approximates the state-dependent linear operator ($\hat{A}_a a^n$, cross-mode coupling).

    2.  **FFN Branch ($\Delta a_{ffn}$):** Approximates the intrinsic non-linear evolution ($I a^n$, self-evolution).

    3.  **BC Branch ($\Delta a_{bc}$):** Explicitly approximates the boundary forcing term ($\Phi^T U_B^{n+1}$).


**4. Performance on Hyperbolic vs. Feedback Systems (Table 2)**

Reviewers noted that BAROM is not always the SOTA on standard hyperbolic PDEs (Table 2).

* **Response:** We acknowledge that for standard problems with simple boundaries, general operators like GNOT or Unisolver are highly effective. However, BAROM is a **specialized architecture** designed for *stability under coupled feedback*.
* The key result is that while BAROM is competitive on standard tasks, it **dominates** on the complex feedback tasks (Table 1)—reducing MSE by ~80% in some cases—where general operators fail due to the lack of explicit boundary forcing mechanisms. This trade-off is intentional: we prioritize robustness in coupled control loops over marginal gains in standard benchmarks.

---

### Meta-Review · Area_Chair_7kwY · 2026-01-07

**Summary:**

The reviewers raised a number of concerns. The main ones are as follows:

1. Limited novelty (reviewer 1kzQ)

2. BAROM_ExpBC uses next-step boundary information while baselines don't, which seems unfair (reviewer 1kzQ).

3. Baseline tuning (reviewer 1kzQ) and missing baselines (reviewer AhyR).

4. Unclear computational efficiency claims (reviewer 1kzQ and AhyR).

5. Relevant ablations missing (reviewer LjNi, f1F2)

6. Lack of comparison with traditional or physics-based reduced-order methods (such as POD-Galerkin) (reviewer LjNi)

7. In a few instances on non-feedback systems (Table 2), BAROM does not uniformly rank first, with methods like GNOT or UNISOLVER performing best on some metrics, but the authors do not analyze the underlying reasons (reviewer LjNi). Reviewer f1F2 also note the over-claimed performance over the hyperbolic PDEs.

8. Presentation (reviewer f1F2)

**Reviewer Concerns:**

After the rebuttal, I think concern 2, 4, 6, 8 are mostly addressed. However, in my opinion, the novelty is still limited and the paper needs more thorough ablation study, better tuning of the baseline methods, and the presentation could be further improved. Therefore, at the current stage I cannot recommend acceptance.

**Reviewer Scores:**

The reviewers' scores will likely remain the same.

---

### Decision · Program_Chairs · 2026-01-26

Reject